# What makes unlearning hard and what to do about it

**Kairan Zhao**[*]
University of Warwick

**Meghdad Kurmanji**
University of Cambridge

**George-Octavian Barbulescu**
University of Warwick

**Eleni Triantafillou**[†]
Google DeepMind

**Peter Triantafillou**[†]
University of Warwick

## Abstract

Machine unlearning is the problem of removing the effect of a subset of training data (the "forget set") from a trained model e.g. to comply with users' requests to delete their data, or remove mislabeled, poisoned or otherwise problematic data. With unlearning research still being at its infancy, many fundamental open questions exist: Are there *interpretable* characteristics of forget sets that substantially affect the difficulty of the problem? How do these characteristics affect different state-of-the-art algorithms? We present the first investigation into these questions. We identify two key factors affecting unlearning difficulty and the performance of unlearning algorithms. Our evaluation on forget sets that isolate these identified factors reveals previously-unknown behaviours of state-of-the-art algorithms that don't materialize on random forget sets. Based on our insights, we develop a framework coined Refined-Unlearning Meta-algorithm (RUM) that encompasses: (i) *refining* the forget set into homogenized subsets, according to different characteristics; and (ii) a *meta-algorithm* that employs existing algorithms to unlearn each subset and finally delivers a model that has unlearned the overall forget set. RUM substantially improves top-performing unlearning algorithms. Overall, we view our work as an important step in deepening our scientific understanding of unlearning and revealing new pathways to improving the state-of-the-art. [‡]

## 1 Introduction

Deep learning models have generated impressive success stories recently by leveraging increasingly large and data-hungry neural networks that are also increasingly expensive to train. This trend has led to reusing previously-trained models for a wide range of tasks more than ever before. However, the heavy reliance of deep models on training data, together with the difficulty of removing data from trained models after-the-fact, has exacerbated concerns on perpetuating harmful or outdated information, violating user privacy and other issues. Specifically, deep networks are highly non-convex, making it difficult to trace (and thus attempt to remove) the effect of a given subset of training data on the model weights. We are therefore faced with important technical challenges when it comes to building machine learning pipelines that are performant while efficiently supporting deletion requests. Machine unlearning [29] is a growing field that aims to address this important issue.

While unlearning is receiving increasing attention [34, 33], it is still a young area of research and the factors affecting the success of different approaches remain poorly-understood. Understanding

---

[*]Correspondence to Kairan.Zhao@warwick.ac.uk

[†]Equal senior contribution

[‡]Code is available at: `https://github.com/kairanzhao/RUM`

what makes an unlearning problem easy or hard is crucial for several reasons. First, knowledge of behaviours of unlearning algorithms on different types of forgetting requests may inform which unlearning method to choose for a given request. In fact, for some requests it may be that all current methods are inadequate, suggesting that one should pay the cost of retraining from scratch rather than opting for "approximate unlearning" that imperfectly removes information after-the-fact. Further, deepening our understanding of unlearning can illuminate pathways for improving both unlearning algorithms as well as evaluation protocols by focusing on relevant factors that affect difficulty.

To this end, we present the first investigation into different factors that characterize the difficulty of an unlearning problem. We find that the unlearning problem becomes harder i) the more entangled the retain and forget sets are and ii) the more memorized the forget set examples are. Our investigation reveals that different unlearning algorithms suffer disproportionately as the difficulty level increases and surfaces previously-unknown behaviours and failure modes of state-of-the-art unlearning algorithms. Inspired by our findings, we propose a Refined-Unlearning Meta-algorithm (RUM) for improving unlearning pipelines. RUM contains two steps: i) a refinement procedure that divides the given forget set into subsets that are homogeneous with respect to relevant factors that influence algorithms' behaviours, and ii) a meta-algorithm that dictates how to unlearn each of those subsets and compose the resulting models to arrive at one that has unlearned the entire forget set. Our thorough investigation shows that RUM boosts unlearning performance of several state-of-the-art algorithms and addresses issues that our investigation of unlearning difficulty has uncovered.

## 2 Preliminaries

### 2.1 Unlearning problem formulation

Let $\theta^o = \mathcal{A}(\mathcal{D}_{train})$ be the weights obtained by applying a training algorithm $\mathcal{A}$ on a training dataset $\mathcal{D}_{train}$. We will refer to $\theta^o$ as the "original model". Further, let $\mathcal{S} \subseteq \mathcal{D}_{train}$ denote a subset of the training data referred to as the "forget set". For convenience, we will refer to its complement as the "retain set" $\mathcal{R} = \mathcal{D}_{train} \setminus \mathcal{S}$. Informally, the goal of an unlearning algorithm $\mathcal{U}$ is to utilize $\theta^o$, $\mathcal{S}$ and $\mathcal{R}$ to produce an unlearned model $\theta^u = \mathcal{U}(\theta^o, \mathcal{S}, \mathcal{R})$ from which the influence of $\mathcal{S}$ is removed.

This idea has been formalized by considering the distributional similarity between the model $\theta^u$ produced by $\mathcal{U}$ and the model $\theta^r$ produced by the optimal unlearning approach: retraining from scratch on an adjusted training dataset that excludes the forget set: $\theta^r = \mathcal{A}(\mathcal{D}_{train} \setminus S)$. Note that we refer to distributions here since rerunning $\mathcal{U}$ and $\mathcal{A}$ with different random seeds (that control e.g. the initialization and the order of mini-batches) will lead to slightly different weights. The ideal unlearning algorithm then, according to this viewpoint, is one that yields the same distribution of weights as retraining from scratch. Of course, for an unlearning algorithm to be practical, we would additionally desire it to be significantly more computationally efficient than retraining the model.

The following definition, borrowed from [33] (and similar to [14]) formalizes this idea in a framework inspired by differential privacy [8].

**Definition 2.1. Unlearning.** An unlearning algorithm $\mathcal{U}$ is an $(\epsilon, \delta)$-*unlearner* (for $\mathcal{A}$, $\mathcal{D}_{train}$ and $\mathcal{S}$) if the distributions of $\mathcal{A}(\mathcal{D}_{train} \setminus S)$ and $\mathcal{U}(\theta^o, \mathcal{S}, \mathcal{D}_{train} \setminus S)$ are $(\epsilon, \delta)$-close.

where we say two distributions $\mu, \nu$ are $(\epsilon, \delta)$-*close* if $\mu(B) \leq e^\epsilon \nu(B) + \delta$ and $\nu(B) \leq e^\epsilon \mu(B) + \delta$ for all measurable events $B$.

According to the above definition, an unlearning algorithm is said to be *exact unlearning* if it satisfies the above definition for $\epsilon = \delta = 0$, i.e., it yields a distribution of models identical to that of retraining from scratch. For neural networks, the only known exact solutions involve retraining, either naively, or in the context of mixtures where one can retrain only a subset of models affected by the deletion request [3]. These approaches unfortunately are inefficient; in the worst-case, even clever schemes suffer inefficiency similar to naive retraining, and may also yield poorer performance. To address this, a plethora of *approximate unlearning* algorithms have been recently proposed, whose $\epsilon$ and $\delta$ values aren't known in general, but are substantially more efficient and may have higher utility.

**Evaluating approximate unlearning.** Since the success of (most) approximate unlearning algorithms cannot be proved within tight $(\epsilon, \delta)$ bounds, the community has considered various empirical measurements of success, guided by three desiderata: 1) good forgetting quality, 2) high utility, and 3) efficiency. An unlearning algorithm thus is faced with a complex balancing act, as there are

well-known trade-offs both between forgetting quality and utility, as well as forgetting quality and efficiency, and a good unlearning metric should capture these nuances.

Utility and efficiency are straightforward to measure, and, in the context of classifiers, can be represented by the accuracy on the retain and test sets, and time in seconds, respectively. Measuring forgetting quality, on the other hand, is more complex and several proxies have been proposed. The simplest one is to inspect the accuracy on the forget set, with the goal of matching the accuracy on the forget set that would have been obtained by retraining from scratch. Alternatively, inspired from privacy literature [4, 28], *Membership Inference Attacks (MIAs)* have been adopted by the unlearning community [25, 26, 20] to measure forgetting quality. In essence, an MIA is designed to infer from the model's characteristics (e.g. loss, confidence) whether a data point has been used in training, and then unlearned, versus was never trained on in the first place. Intuitively, the failure of an attacker to tell apart unlearned examples from never-seen examples marks a success for the unlearning algorithm in terms of this metric. We will consider both of these proxies in our experimental investigation.

To holistically evaluate an unlearning algorithm, we desire a single metric that captures both forgetting quality and utility. We will later introduce a "tug-of-war" metric for this purpose, inspired by [33].

## 2.2 Memorization

Deep neural networks are known to "memorize" (a subset of) their training data, with a recent theory showing that label memorization is in fact necessary for achieving close-to-optimal generalization error in classifiers [11] when the data distribution is long-tailed.

**Definition 2.2. Memorization score [11].** The *memorization score* for an example $i \in \mathcal{D}$, with respect to a training dataset $\mathcal{D}$ and training algorithm $\mathcal{A}$ is

$$\text{mem}(\mathcal{A}, \mathcal{D}, i) = \Pr_{f \sim \mathcal{A}(\mathcal{D})}[f(x_i) = y_i] - \Pr_{f \sim \mathcal{A}(\mathcal{D} \setminus i)}[f(x_i) = y_i] \tag{1}$$

where $x_i$ and $y_i$ are the feature and label, respectively, of example $i$.

The first term in the above equation considers models trained on all of $\mathcal{D}$ whereas the second term considers models trained on $\mathcal{D}$ excluding example $i$. Intuitively, the memorization score for an example $i$ is high if including it in training yields a different distribution of predictions on that example than excluding it from training would have. Recent works [11, 12, 23] identify atypical examples or outliers of the data distribution as examples that are more highly memorized: if an example has a noisy or incorrect label, the model is required to memorize it in order to predict it correctly.

## 3 Related Work

**Approximate unlearning algorithms.** A plethora of algorithms have been proposed that aim to identify effective data scrubbing procedures post-training. We now describe representative methods.

**Fine-tune** [36, 15] relies on catastrophic forgetting to diminish the confidence of the original model $\theta^o$ on $\mathcal{S}$. Catastrophic forgetting is induced by simply fine-tuning on the retain set $\mathcal{D}_{train} \setminus \mathcal{S}$. On the other hand, **NegGrad** [15, 17, 31] instead directly maximizes the loss on $\mathcal{S}$. This approach has been found empirically to cause a large drop in the utility of the model. To address this, **NegGrad+** [25] combines fine-tuning and gradient ascent, by jointly minimizing the loss function on the retain set, and maximizing the loss function with respect to the forget set. **SCRUB**, proposed by the same authors as NegGrad+, extends the contrastive learning behind NegGrad+ by framing it as a student-teacher problem. Concretely, SCRUB is a bi-optimization algorithm, where the student aims to mimic the teacher's behaviour on $\mathcal{R}$ and to disobey the teacher's output with respect to $\mathcal{S}$. **L1-sparse** [26] infuses weight "sparsity" into the unlearning algorithm by fine-tuning on the retain set with an L1-penalty, drawing inspiration from the model pruning literature [13, 27]. **Influence Unlearning** [21, 24] arrives at the important model's weights by estimating how removing a data point affects $\theta^o$ via influence functions [7], and draws connections to $(\epsilon, \delta)$-*forgetting* [35, 19].

A different line of work is "relabelling-based" methods that trick the model to learn new labels for $\mathcal{S}$. This can be achieved by finetuning the model with respect to a dataset $\mathcal{D}_{relabel} = (X_\mathcal{S}, Y)$, where $X_\mathcal{S}$ are the features and labels $Y$ are sampled from a prior distribution of the label space. **Saliency Unlearning (SalUn)** [10] learns $\mathcal{D}_{relabel}$ by optimising only the *salient* parameters of the model.

Concretely, the authors argue that the model's weights $\theta^o$ can be decomposed into salient weights and "intact" model weights, by investigating the weight space with respect to the forget set $S$ ala [30, 1].

**Difficulty of Unlearning.** The closest research to ours is the contemporaneous work of [9], where the authors study *adversarial unlearning* cases, i.e. "worst-case" forget sets. [9] arrives at difficult forget sets by solving a bi-level optimization based on fine-tuning (i.e. catastrophic forgetting). Instead, we arrive at difficult partitions through the lens of *interpretable* factors: the degree of entanglement between the retain and forget set and memorization. While the primary aim of [9] is to construct more pessimistic evaluation benchmarks, our primary aim is to deepen our understanding of unlearning problems and of the behaviour of state-of-the-art algorithms when operating on forget set of different identified characteristics, ultimately improving unlearning pipelines.

**Catastrophic forgetting, atypical examples and privacy.** [22] and [32] study catastrophic forgetting during training. [22] finds that, when training on large datasets, examples that were only seen early in training may enjoy better privacy, in terms of MIAs and extraction attacks, compared to examples seen recently. [32] investigate "forgetting events", where an example that was previously correctly predicted becomes incorrectly predicted later in training. They find that examples with noisy labels witness a larger number of these forgetting events. Further, [5] find that models trained with Differential Privacy (DP) find it primarily hard to correctly predict atypical examples. We build on this literature by studying the difficulty of unlearning after-the-fact, rather than (passive) forgetting during training and draw connections to memorization, a notion closely related to atypicality in the data distribution.

# 4 What Makes Unlearning Hard?

In this section, we identify and empirically examine two factors that affect the difficulty of unlearning. Before diving in, we first define a simple proxy for unlearning difficulty that we will use in this section. Our goal is to capture the difficulty of performing the "balancing act" of forgetting $\mathcal{S}$ while retaining the ability to perform well on $\mathcal{R}$ and generalize well to the test set. We propose a metric to capture this "tug-of-war" (ToW) using the relative difference between the accuracies of the unlearned and the retrained model on the forget, retain and test sets, in a manner inspired by [33].

$$\text{ToW}(\theta^u, \theta^r, \mathcal{S}, \mathcal{R}, \mathcal{D}_{test}) = (1 - \text{da}(\theta^u, \theta^r, \mathcal{S})) \cdot (1 - \text{da}(\theta^u, \theta^r, \mathcal{R})) \cdot (1 - \text{da}(\theta^u, \theta^r, \mathcal{D}_{test}))$$

where $\text{a}(\theta, \mathcal{D}) = \frac{1}{|\mathcal{D}|} \sum_{(x,y) \in \mathcal{D}} [f(x; \theta) = y]$ is the accuracy on $\mathcal{D}$ of a model $f$ parameterized by $\theta$ and $\text{da}(\theta^u, \theta^r, \mathcal{D}) = |\text{a}(\theta^u, \mathcal{D}) - \text{a}(\theta^r, \mathcal{D})|$ is the absolute difference between the accuracy of models $\theta^u$ and $\theta^r$ on $\mathcal{D}$. Therefore, ToW rewards unlearned models that match the accuracy of the retrained-from-scratch model, on each of the forget, retain, and test sets. ToW ranges from 0 to 1, with higher values associated with better unlearning.

## 4.1 The more entangled the forget and retain sets are, the harder unlearning becomes

Prior research (e.g., by Feldman et al [11] and Carlini et al [5]) has tried to identify prototypical (or atypical) examples and their impact on learning. Primarily, this depended on the position of *examples* within the overall *data space distribution*. In contrast, here we focus on the *embedding space*, as unlearning depends heavily on how the model has learned to represent training data. Furthermore, instead of looking at isolated examples, we delve into how "entangled" the retain and forget sets are in embedding space. We hypothesize that higher "entanglement" leads to harder unlearning: if the two sets are highly entangled, attempting to erase $\mathcal{S}$ will cause accidentally erasing $\mathcal{R}$ too.

We propose to measure entanglement between the retain and forget sets via the below *Entanglement Score* (ES), inspired by a measure previously introduced in [16] to study learned representations.

$$\text{ES}(\mathcal{R}, \mathcal{S}; \theta^o) = \frac{\frac{1}{|\mathcal{R}|} \sum_{i \in \mathcal{R}} (\phi_i - \mu_{\mathcal{R}})^2 + \frac{1}{|\mathcal{S}|} \sum_{j \in \mathcal{S}} (\phi_j - \mu_{\mathcal{S}})^2}{\frac{1}{2} \left( (\mu_{\mathcal{R}} - \mu)^2 + (\mu_{\mathcal{S}} - \mu)^2 \right)} \tag{2}$$

where $\phi_i = g(x_i; \theta^o)$ is the embedding of example $x_i$ according to the "original model" $f$, parameterized by $\theta^o$; where $g$ denotes the forward pass through $f$ up till the penultimate layer, i.e. excluding the

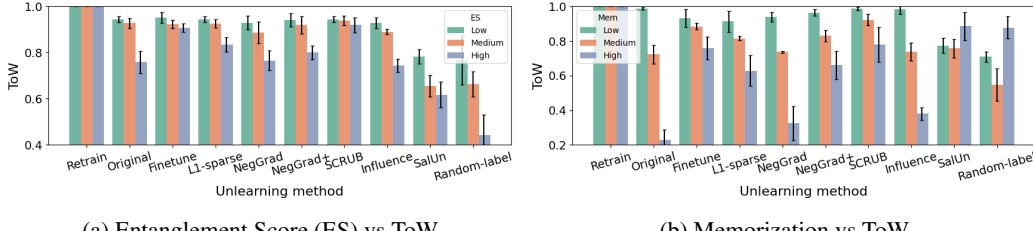

(a) Entanglement Score (ES) vs ToW                    (b) Memorization vs ToW

Figure 1: Uncovering two factors that affect unlearning difficulty according to ToW (where higher is better). **Left:** the more entangled the retain and forget sets are in the embedding space, the harder it is to unlearn. **Right:** the less memorized a forget set is (thus having influenced the model less), the easier it is to unlearn (for most algorithms). Error bars correspond to 95% confidence intervals from running each algorithm 3 times (6 times for relabelling-based that had higher variance).

classifier layer. Further, $\mu_{\mathcal{R}} = \frac{1}{|\mathcal{R}|} \sum_{i \in \mathcal{R}} \phi_i$ is the mean embedding of the retain set, and analogously, $\mu_{\mathcal{S}}$ the mean embedding of the forget set, while $\mu$ is the mean embedding over all of $\mathcal{D}_{train} = \mathcal{R} \cup \mathcal{S}$.

Intuitively, ES measures entanglement between $\mathcal{S}$ and $\mathcal{R}$ in the embedding space of the original model (before unlearning begins). The numerator measures intra-class variance, capturing the tightness of each of those two sets, independently, while the denominator measures inter-class variance between those two sets. Higher ES score corresponds to higher entanglement in the embedding space.

Our investigation hinges on generating three different forget/retain partitions with different degrees of entanglement: low, medium, and high. But, Equation (2) does not directly suggest a procedure for generating retain/forget partitions with a desired ES score. Hence, we achieved this indirectly using a proxy. Specifically, let $d(i, \mu; \theta^o) = ||\phi_i - \mu||^2$ denote the l2-distance in the original model's embedding space between example $i$ and centroid $\mu$, as defined above. We compute this distance for each example in $\mathcal{D}_{train}$ and sort those examples according to their $d$-values. We then form each forget set to contain a contiguous subset of examples from different ranges of that sorted list. We find that this procedure allows us to construct retain/forget partitions of varying ES values. The ES values for our low, medium and high partitions are 309.94±98.56, 1076.99±78.64, 1612.21±110.82 for CIFAR-10, and 963.82±113.53, 2831.24±558.63, and 3876.90±426.92 for CIFAR-100. We include details of this procedure in the Section A.3, along with visualizations and Maximum Mean Discrepancy (MMD) analysis to further validate ES, confirming that the degree of retain/forget entanglement aligns with the computed scores. We experiment with four dataset/architecture settings: CIFAR-10/ResNet-18, CIFAR-100/ResNet-50, Tiny-ImageNet/ResNet-18 and Tiny-ImageNet/VGG-16, using $|S| = 3000$. Refer to Section A.2 for implementation details and Section A.8 for more detailed results on all datasets.

We observe from Figure 1a that it is harder to unlearn when the retain and forget sets are more entangled: all unlearning algorithms have poorer performance for highly-entangled vs lower-entangled settings. Further, we notice that different unlearning algorithms suffer disproportionately as the entanglement increases. Notably, methods based on relabelling (SalUn and Random-label) perform very poorly when the entanglement is high. We hypothesize that this is because, if two examples $i$ and $j$ are close neighbours in embedding space, with $i$ in the forget set and $j$ in the retain set, forcing example $i$ to be confidently predicted as an incorrect class (as relabelling algorithms do) will also cause $j$ to be predicted as that incorrect class, too, thus causing a drop in retain accuracy, which is captured by ToW. This effect will be less pronounced if $i$ and $j$ are far from each other.

### 4.2 The more memorized the forget examples are, the harder unlearning becomes

Feldman et al [11] have already established that models must memorize some atypical examples in order to perform well. Further, prior literature has also established that noisy examples (that are more likely to be memorized) witness more "forgetting events" during training (their predicted label flips to an incorrect one) [32] and that models trained with Differential Privacy (DP), a procedure where noise is added to the gradients (making it harder to memorize), find it primarily hard to correctly predict atypical examples [5]. In this section, we build upon these prior insights by investigating the connection between the degree of memorization of the forget set and difficulty of unlearning.

Let's begin by inspecting Definition 2.2: if an example is not really memorized, the predictions of the model on that example will not change much whether the example was included in training or not. This implies that even the original model (no unlearning) is similar to retrain-from-scratch in terms of predictions on those examples, making unlearning unnecessary or trivial. On the other hand, for highly-memorized examples, the predictions between the original and retrained models will differ significantly, implying that an unlearning algorithm has "more work" to do to turn the original model into one that resembles the retrained one. We now investigate how the level of memorization of the forget set affects the behaviour of state-of-the-art unlearning algorithms. We hypothesize, based on our above intuition, that unlearning is easier when the forget set contains less-memorized examples.

To investigate this, we first compute the memorization score $\text{mem}(\mathcal{A}, \mathcal{D}_{train}, i)$ of each example $i \in \mathcal{D}_{train}$ and we sort all examples according to their scores. We then use that sorted list to create three different forget sets, corresponding to the lowest $N$ scores ("low-mem"), the highest $N$ ("high-mem"), and the $N$ that are nearest to 0.5, i.e. the midpoint of the range of memorization scores ("medium-mem"), where $N = 3000$. We then apply different unlearning algorithms on each of these forget sets and compute ToW. We perform this experiment on CIFAR-10 using ResNet-18 and on CIFAR-100 using ResNet-50. Refer to Section A.2 for implementation details.

We first emphasize two key sets of conclusions. First, in terms of ToW, Figure 1b shows that, indeed, for most algorithms, the lower the memorization level of the forget set, the easier the problem is. In line with our prior discussion, even the original model performs well on "low-mem", but performs very poorly on "high-mem". Interestingly though, the two relabelling-based algorithms (SalUn and Random-label) follow an inverse trend: they perform better for higher-memorized forget sets. Second, breaking down ToW into its parts, we find interesting trends in terms of the forget set accuracy, in Figure 12. Specifically, for several unlearning algorithms, the forget accuracy for "low-mem" is still very high after unlearning them, as the model can infer the correct labels for such examples even when they weren't included in training; this follows directly by Definition 2.2 if unlearning is done by retraining, and is shown here for the first time for approximate unlearning algorithms. On the other hand, we find that, for "high-mem", different unlearning algorithms can (to varying degrees) cause the forget set accuracy to drop substantially; this is consistent with both [32] and [5], but shown here for the first time for approximate unlearning algorithms. Notably, we find that relabelling-based algorithms cause a larger drop in the accuracy of the forget set, relative to other approaches. This benefits ToW in the case of "high-mem" forget sets, where retraining has poor accuracy on this set (so they get rewarded by matching it), but it hurts on "low-mem", since it causes a large discrepancy from retraining, which has high accuracy on this set (since it makes similar predictions to the original model on this set, by definition, and the original model has high accuracy on all of $\mathcal{D}_{train}$).

Overall, we have presented the first investigation into the behaviour of unlearning algorithms applied on forget sets of different degrees of memorization. A key finding is that different algorithms outshine others for different forget sets. Most notable is the failure of relabelling-based algorithms on the "low-mem" forget set, which is easy for other algorithms and, in fact, even no unlearning in that case might be an acceptable solution. We intuit that this is due to their aggressive unlearning strategy yielding "overforgetting" (producing a forget set accuracy that is lower than that of retraining from scratch) as discussed above. Furthermore, we observe that different unlearning algorithms work best for the "low-mem", "medium-mem" and "high-mem" forget sets. Concretely, from Figure 1b we note that Finetune is best for "medium-mem", SalUn is best for "high-mem", and a number of algorithms are top-performers for "low-mem" (including no unlearning). This reveals a possible pathway for improving unlearning based on using different algorithms for different forget sets. So, how can one build on these insights to further improve unlearning algorithms performance?

## 5 Refined-Unlearning Meta-algorithm (RUM) for Improved Unlearning

Previously, we observed that unlearning algorithms have different behaviours on forget sets with different properties. For example, while "low mem" forget sets are almost trivial to unlearn (and even doing nothing may be acceptable), SalUn and Random-label perform poorly on them. On the other hand, SalUn and Random-label evidently outperform other unlearning algorithms on "high mem". These observations suggest that the optimal unlearning algorithm to use is dependent on the properties of the forget set. One could therefore pick the best unlearning algorithm for each unlearning request, based on these factors. However, in practical scenarios, forget sets may be distributed differently than in our preliminary experiments, that were designed to cleanly separate different factors of interest.

Indeed, real-world forget sets would likely contain a mixture of examples from different modes of the data distribution, some rare or highly-memorized while others common and not memorized at all. So, what can be done about these expected heterogeneous forget sets? How can our insights above be leveraged to improve unlearning for such cases?

To address this, we first propose a *refinement procedure* that divides forget sets into homogeneous subsets (with respect to the factors that we have found to affect the difficulty of unlearning and behaviours of existing algorithms). Second, we propose to utilize a pool of state-of-the-art algorithms to unlearn different subsets. Put together, we propose a Refined-Unlearning Meta-algorithm (RUM), comprised of two steps: 1) Refinement and 2) Meta-unlearning. Figure 2 overviews RUM.

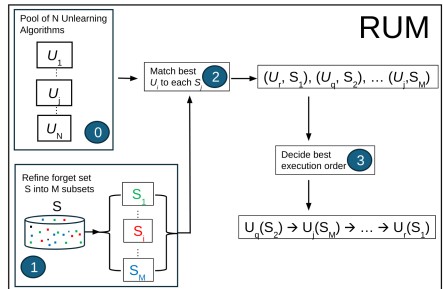

Figure 2: Overview of RUM.

**Step 1: Refinement.** We introduce a function $\mathcal{F}$ that partitions the forget set $S$ into $K$ subsets: $\{S_i\}_{i=1}^{K} = \mathcal{F}(\mathcal{S})$ such that each forget set example appears in exactly one such subset. The intention of $\mathcal{F}$ is to generate homogeneous subsets w.r.t factor(s) that affect difficulty / algorithm's behaviours.

**Step 2: Meta-Unlearning.** Having obtained the subsets $\{S_i\}_{i=1}^{K}$ of $\mathcal{S}$, we now require a "meta-algorithm" $\mathcal{M}$ that dictates how to perform the individual unlearning requests and how to compose the resulting unlearned models to arrive to a model that has unlearned all of $\mathcal{S}$. In this work, we focus on meta-algorithms that tackle unlearning of subsets in a sequence, leaving other designs for future work. It therefore remains for our "meta-algorithm" to decide: i) what unlearning algorithm to apply for each subset, ii) what order should the unlearning requests be executed in.

More concretely, we assume access to a pool of existing unlearning algorithms $\mathcal{U}_1 \ldots \mathcal{U}_N$, like the ones described in related work, for instance. Let $\mathcal{M}^{\mathcal{U}}(S_i)$ denote a procedure that takes as input a subset $S_i$ and returns an unlearning algorithm $\mathcal{U} \in \{\mathcal{U}_1 \ldots \mathcal{U}_N\}$ that will be used for that subset. This selection can be done by leveraging insights such as those in Section 4. Further, let $\mathcal{M}^{\mathcal{O}}$ denote a procedure that takes as input the $K$ subsets of $\mathcal{S}$ and returns a sorted list $S' = \mathcal{M}^{\mathcal{O}}(\mathcal{F}(\mathcal{S}))$ containing the K subsets in the desired order of execution.

Given the above ingredients, RUM proceeds by executing $K$ unlearning requests in a sequence, with step $i$ of that sequence corresponding to unlearning subset $S'[i]$ by applying $\mathcal{U}_i(\theta^o, S'[i], \mathcal{R}_i) = \theta_i^u$, where $\theta_i^u$ denotes the unlearned model up to step $i$ and $\mathcal{U}_i = \mathcal{M}^{\mathcal{U}}(S'[i])$ and $\mathcal{R}_i = \mathcal{R} \cup \{S'[i+1], \ldots S'[K]\}$ is the retain set for step $i$, containing $\mathcal{R}$ as well as all other subsets of $\mathcal{S}$ that have not yet been unlearned in the sequence so far. We finally return the unlearned model of the last step $\theta_K^u$.

Our RUM framework is meant as an analysis framework, surfacing new problems to be solved and offering new pathways into future state-of-the-art algorithms. Nonetheless, we contribute below specific top-performing RUM instantiations, with specific choices for $\mathcal{F}$ and for $\mathcal{M}$.

# 6 Experimenting with RUM flavours

We now present RUM instantiations using a refinement strategy based on memorization scores and experimental evaluations answering the following questions: **Q1**: How useful is refinement alone? That is, for a given unlearning algorithm $\mathcal{U}$, does applying $\mathcal{U}$ sequentially on the $K$ homogeneous subsets of $\mathcal{S}$ outperform applying $\mathcal{U}$ once on all of $\mathcal{S}$? **Q2**: Can we obtain further gains by additionally selecting the best-performing unlearning algorithm for each forget set subset? **Q3**: Are there interpetable factors behind the boost obtained by sequential unlearning of homogeneous subsets?

**Experimental setup** We experiment with a refinement strategy based on memorization scores where $K = 3$. Specifically, we study unlearning a forget set $\mathcal{S}$ that is the union of the three sets containing the $N$ lowest, the $N$ closest to 0.5, and the $N$ highest memorized examples in the dataset,

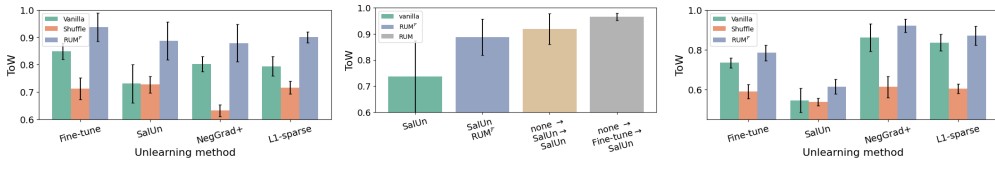

| (a) CIFAR-10 with ResNet-18 | (b) CIFAR-10 with ResNet-18 | (c) CIFAR-100 with ResNet-50 |

Figure 3: From subplots a and c, we observe that **RUM$^{\mathcal{F}}$ improves each unlearning algorithm**. Vanilla corresponds to unlearning $\mathcal{S}$ in one go, whereas Shuffle and RUM$^{\mathcal{F}}$ operate sequentially on 3 subsets of $\mathcal{S}$. In the case of RUM$^{\mathcal{F}}$, the 3 subsets are the result of applying $\mathcal{F}$ and the order is low $\rightarrow$ medium $\rightarrow$ high, whereas Shuffle uses equal-sized random subsets, serving as a control experiment. Further, from subplot b, we observe that **full RUM, equipped with the best algorithm for each subset** (do nothing $\rightarrow$ Fine-tune $\rightarrow$ SalUn), **yields the overall best results** (note: in CIFAR-100, NegGrad+ is the best algorithm, so full RUM corresponds to the RUM$^{\mathcal{F}}$ variant of NegGrad+).

where $N = 1000$, making the overall size of the forget set 3000. We conduct experiments on two different datasets: CIFAR-10 and CIFAR-100, using ResNet-18 and ResNet-50, respectively. We evaluate unlearning algorithms both in terms of ToW, as before, and using a commonly-used MIA [10, 26] that is a binary classifier trained to separate $\mathcal{R}$ from $\mathcal{D}_{test}$ and then queried on examples from $\mathcal{S}$. Following previous work, we report as "MIA" the fraction of examples from $\mathcal{S}$ that were predicted to be held-out. The goal is to match the "MIA" score of retrain-from-scratch. For convenience, we report the "MIA gap": the absolute difference of the MIA score of unlearning from the MIA score of retrain-from-scratch (lower is better). Further details on MIA setup are in Section A.4. We also introduce a new metric "ToW-MIA", which mirrors ToW but replaces its "forget quality" component (i.e., its accuracy on $\mathcal{S}$) with the MIA score. Results for ToW-MIA are presented in Section A.4 and A.8 and largely follow the same trends. To gain a finer understanding of the differences between the unlearned and retrained models, we additionally analyze the number of examples where their predictions disagree (see Section A.7). We also consider and analyze the effect of two different orderings: i) low $\rightarrow$ medium $\rightarrow$ high and i) high $\rightarrow$ medium $\rightarrow$ low.

**How useful is refinement alone?** To investigate this, we selected a subset of highest-performing algorithms and apply each algorithm $\mathcal{U}$ in three ways: i) "vanilla", i.e. applying $\mathcal{U}(\theta^o, \mathcal{S}, \mathcal{R})$ as usual, ii) "shuffle" where $\mathcal{U}$ is applied sequentially on three equal-sized subsets of $\mathcal{S}$ that were determined randomly, and iii) RUM$^{\mathcal{F}}$, where $\mathcal{U}$ is applied sequentially on the subsets obtained by $\mathcal{F}(\mathcal{S})$, i.e. utilizing only the refinement step of RUM and applying the same $\mathcal{U}$ on each subset. We include "shuffle" as a control experiment, so that any gain of RUM$^{\mathcal{F}}$ over shuffle is due to homogenization rather than simply reducing the size of the forget set or other effects of sequential unlearning. We observe from Figure 3 and Tables 16a and 16b that, for four different unlearning algorithms and on two different datasets, RUM$^{\mathcal{F}}$ significantly outperforms "vanilla" and "shuffle", indicating that operating sequentially on homogenized subsets according to memorization can boost the performance of unlearning algorithms. Interestingly, in most cases, "shuffle" actually performs worse than "vanilla", indicating that the difficulty of the problem may increase rather than decrease given a poor refinement strategy.

|  | CIFAR-10 | | CIFAR-100 | |
|---|---|---|---|---|
|  | ToW ($\uparrow$) | MIA gap ($\downarrow$) | ToW ($\uparrow$) | MIA gap ($\downarrow$) |
| Retrain | 1.000±0.000 | 0.000 | 1.000±0.000 | 0.000 |
| Fine-tune vanilla | 0.849±0.030 | 0.120 | 0.734±0.025 | 0.139 |
| Fine-tune shuffle | 0.712±0.040 | 0.098 | 0.589±0.036 | 0.345 |
| Fine-tune RUM$^{\mathcal{F}}$ | **0.937±0.052** | 0.099 | **0.784±0.040** | 0.093 |
| L1-sparse vanilla | 0.794±0.035 | 0.175 | 0.824±0.011 | 0.089 |
| L1-sparse shuffle | 0.716±0.023 | 0.257 | 0.604±0.023 | 0.353 |
| L1-sparse RUM$^{\mathcal{F}}$ | **0.900±0.020** | 0.072 | **0.883±0.046** | 0.033 |
| NegGrad+ vanilla | 0.802±0.028 | 0.230 | 0.861±0.069 | 0.159 |
| NegGrad+ shuffle | 0.632±0.022 | 0.520 | 0.613±0.054 | 0.417 |
| NegGrad+ RUM$^{\mathcal{F}}$ | **0.879±0.068** | 0.134 | **0.921±0.034** | 0.059 |
| SalUn vanilla | 0.731±0.070 | 0.374 | 0.545±0.061 | 0.372 |
| SalUn shuffle | 0.727±0.030 | 0.234 | 0.538±0.019 | 0.237 |
| SalUn RUM$^{\mathcal{F}}$ | **0.887±0.069** | 0.031 | **0.614±0.037** | 0.181 |
| RUM | **0.965±0.014** | 0.034 | **0.921±0.034** | 0.059 |

Table 1: Each algorithm $\mathcal{U}$ is applied in three ways: i) in one-go ("vanilla"), ii) on a random partition of $\mathcal{S}$ into 3 equal-sized subsets, sequentially ("shuffle"), and iii) on three equal-sized subsets obtained by $\mathcal{F}$ in low $\rightarrow$ med $\rightarrow$ high order ("RUM$^{\mathcal{F}}$"). In last row, RUM additionally chooses the best algorithm for each subset: none $\rightarrow$ Fine-tune $\rightarrow$ SalUn in CIFAR-10 and NegGrad+ RUM$^{\mathcal{F}}$ in CIFAR-100.

**Can we further boost performance by per-subset algorithm selection?** To answer this, we leverage our findings from Section 4 to identify the best unlearning algorithm for each of the "low-

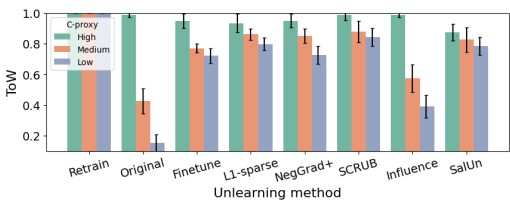
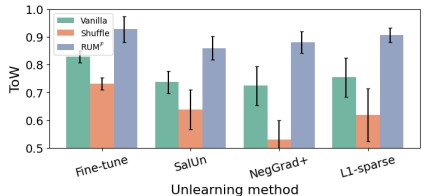

(a) C-proxy vs ToW

(b) RUM$^{\mathcal{F}}$ on CIFAR-10 using C-proxy

Figure 4: Replacing mem scores with the efficient C-proxy yields similar trends and performance gains, carving a path for practical deployment of RUM. **Left:** forget sets with lower C-proxy values (i.e., higher mem scores, since C-proxy and memorization are negatively correlated) are harder to unlearn, consistent with the trend in Figure 1b. **Right:** RUM$^{\mathcal{F}}$ using C-proxy in the refinement step enhances unlearning performance across algorithms, comparable to using the memorization score in Figure 3a. Error bars correspond to 95% confidence intervals, with each algorithm run 3 times.

mem", "medium-mem" and "high-mem" scenarios. On CIFAR-10, this corresponds to doing nothing for low-mem (i.e. using the original model directly), using Fine-tune for medium-mem and SalUn for high-mem (based on Figure 1b). From Figure 3b (and Table 16a), we observe that we get the overall best results by far, both in terms of ToW and MIA, by applying RUM with "nothing → Fine-tune → SalUn", demonstrating the value of incorporating our insights into unlearning pipelines. In fact, lets revisit our previous observation that SalUn and Random-Label perform uncharacteristically poorly on the "low-mem" forget set. In line with our hypothesis, we notice that "nothing → SalUn → SalUn" outperforms applying SalUn on all three subsets (a.k.a. SalUn RUM$^{\mathcal{F}}$). Note that, for CIFAR-100, the best algorithm for all subsets is NegGrad+, so full RUM corresponds to NegGrad+ RUM$^{\mathcal{F}}$.

**Can we obtain similar performance boosts with compute-efficient proxies?** Given the computational cost of calculating memorization scores, we aim to find a more efficient proxy to enable practical deployment of RUM. We discuss a confidence-based memorization metric, termed "C-proxy", as a proxy for memorization [23, 32]. Section A.6 provides a detailed explanation of the proxy and the complete results across various datasets and architectures. Figure 4a shows that the observed unlearning difficulty pattern—specifically, that forget examples with higher memorization scores (i.e., lower C-proxy values, as they are negatively correlated; see Table 4) are harder to unlearn—remains consistent when using C-proxy, paralleling the results in Figure 1b. This pattern is further confirmed on Tiny-ImageNet with both ResNet-18 and the VGG-16 architectures, as shown in Figure 9. We then examine whether using C-proxy achieves similar performance gains within RUM as the original memorization score. Figure 4b presents results on CIFAR-10 with ResNet-18, using C-proxy in place of memorization score during the refinement step. This analysis is also extended to Tiny-ImageNet with both ResNet-18 and VGG-16 architectures, as shown in Figure 10 and Table 17. Together, these results suggest that C-proxy is a practical and compute-efficient alternative, delivering significant performance gains comparable to those achieved with the memorization score. Detailed analysis of the use and appropriateness of various memorization proxies and their effect on RUM can be found in [37].

**Analysis of sequence dynamics** We report ToW and MIA with different orderings in Tables 16a and 16b. We find that, while ToW is similar for different orderings, MIA can vary. To better understand these dynamics, we inspect the accuracies on $\mathcal{S}$ and its subsets after each step in Figure 5 for SalUn$^{\mathcal{F}}$ on CIFAR-10 and in Figure 13 for NegGrad+$^{\mathcal{F}}$ CIFAR-100. The former reveals why SalUn$^{\mathcal{F}}$ with low → med → high order greatly outperforms vanilla SalUn. Recall that we identified in Section 4.2 that SalUn "overforgets" low-mem examples (its forget accuracy is lower than that of retraining). We observe from Figure 5 that future steps of the sequence neutralize that overforgetting effect on low-mem, leading to better ToW (see Figure 3). Interestingly, in line with our previous insights (Section 4.2), we find (from both Figures 5 and 13) that it is hard to cause the "low-mem" accuracy to become low and stay low. NegGrad+ does not drop it for any order of execution; SalUn drops it in the first ordering, but that drop is later reversed. We leave it to future work to further study these sequential dynamics and their helpful or harmful effects on Tow and MIA. The fact that MIA results differ based on the sequence may tie in with the recently-identified "privacy onion effect" [6].

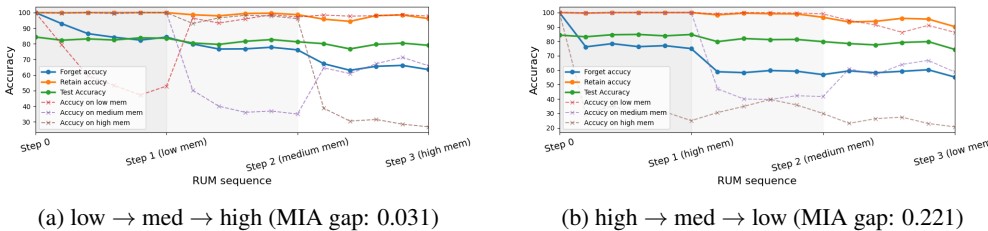

| (a) low $\to$ med $\to$ high (MIA gap: 0.031) | (b) high $\to$ med $\to$ low (MIA gap: 0.221) |

Figure 5: Sequence dynamics for SalUn $\text{RUM}^{\mathcal{F}}$ on CIFAR-10. We report the accuracy on overall $\mathcal{S}$, $\mathcal{R}$ and $\mathcal{D}_{test}$, and subsets of $\mathcal{S}$ after each step. Both orderings yield similar Tow (see Table 16a).

# 7  Discussion and conclusion

We presented the first investigation into interpretable factors that affect the difficulty of unlearning. We found that unlearning gets harder i) the more entangled the forget and retain sets are in embedding space and ii) the more memorized the forget set is. Our investigation led into uncovering previously-unknown behaviours of state-of-the-art algorithms that were not surfaced when considering random forget sets: when do they fail, when do they excel and when do they exhibit different trends from each other. Notably, we discovered that relabelling-based methods suffer disproportionately as the embedding-space entanglement increases, and exhibit a reverse trend compared to other methods in their behaviour for different memorization levels. Armed with these insights, we then proposed the RUM framework surfacing new (sub)problems within unlearning, whose solution may lead to greater performance. Finally, we derived specific instantiations of RUM and analyzed how its different components can improve performance. We found that sequential unlearning of homogenized forget set subsets improves all considered state-of-the-art unlearning algorithms and investigated the dynamics of sequential unlearning to glean insights as to why that is. We also found that we can further boost performance by selecting the best unlearning algorithm per subset.

**Efficiency**   How is this important aspect affected in our sequential framework? We remark that it depends on the unlearning algorithm. For instance, applying Fine-tune three times is much more expensive than applying it once, because Fine-tune performs (at least) one epoch over the entire retain set. But for other algorithms the overall cost does not increase significantly. Further, a key observation from our results is that we can do well by actually *doing nothing* on a subset of the forget set, which can really boost efficiency (especially since the vast majority of examples are "low mem"). Additionally, our results with the C-proxy demonstrate that significant performance improvements in unlearning can be achieved with minimal computational cost, avoiding the heavy expense of computing memorization scores.

**Data-space vs embedding-space outliers**   How does the embedding space entanglement interact with the level of memorization of the forget set? We analyzed this and found in Table 3 that all of our memorization buckets have relatively high ES, indicating that separating out (data-space) outliers in the forget set doesn't lead to lower entanglement between the two sets in the embedding space. We leave it to future work to study the interaction of these factors.

**Limitations and future work**   We hope future work explores other refinement strategies (e.g. for a notion that captures embedding-space entanglement) and investigates privacy implications of sequential RUM, e.g. in terms of the "privacy onion effect" [6]. We also hope to see how our RUM framework can be adopted and adapted for unlearning in LLMs, especially given the findings from the contemporaneous paper [2] where unlearning is performed only on the highest-memorized examples in the forget set (albeit, memorization is defined differently for LLMs). We hope our framework continues to enable progress in understanding and improving unlearning and that our identified factors of difficulty and associated behaviours of existing algorithms continue to improve the state-of-the-art and inform the development of strong evaluation metrics that consider forget sets that vary in terms of relevant identified characteristics.

## 8 Acknowledgements

We thank Vincent Dumoulin and Fabian Pedregosa for valuable conversations and feedback at various stages of the project.

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

# A  Appendix / supplemental material

## A.1  Broader impact

Unlearning research can have profound broader impact in allowing users to request their data to be deleted from models or making models safer or more accurate by removing harmful or outdated information. Our work is exploratory: we identify factors affecting difficulty that are interpretable and can drive improvements to unlearning pipelines. As such, we don't see any direct harmful impact of our work.

## A.2  Implementation details

**Datasets and models**  We use the CIFAR-10, CIFAR-100, and Tiny-ImageNet datasets for our evaluation. CIFAR-10 consists of 60,000 32x32 color images across 10 classes, with 6,000 images per class. CIFAR-100 contains 100 classes, each with 600 32x32 color images. Tiny-ImageNet includes 100,000 64x64 color images across 200 classes, with each class containing 500 training images, 50 validation images, and 50 test images (for a total of 100,000 training, 10,000 validation, and 10,000 test images). For CIFAR-10 and CIFAR-100, the train, validation, and test set sizes are 45,000, 5,000, and 10,000 images, respectively. We use ResNet-18, ResNet-50, and VGG-16 as model architectures. Specifically, we use ResNet-18 for CIFAR-10, ResNet-50 for CIFAR-100, and both ResNet-18 and VGG-16 for Tiny-ImageNet.

**Training details for original models**  We trained four models across different datasets and architectures: ResNet-18 on CIFAR-10, ResNet-50 on CIFAR-100, ResNet-18 on Tiny-ImageNet, and VGG-16 on Tiny-ImageNet. For CIFAR-10, the ResNet-18 model was trained for 30 epochs using the SGD optimizer. The learning rate was initialized at 0.1 and scheduled with cosine decay. For CIFAR-100, the ResNet-50 model was trained for 150 epochs using the SGD optimizer, with the learning rate initialized at 0.1 and decayed by a factor of 0.2 at epochs 60 and 120. For Tiny-ImageNet, the ResNet-18 model was trained for 80 epochs, and the VGG-16 model was trained for 100 epochs, both using the SGD optimizer with an initial learning rate of 0.1 and cosine decay. All models were trained with a weight decay of 0.0005, a momentum of 0.9, and a batch size of 256. Additionally, data augmentations, including random cropping and random horizontal flipping, were applied during training on CIFAR-100 and Tiny-ImageNet. All training was conducted on Nvidia RTX A5000 GPUs.

**Training details for machine unlearning**  To ensure optimal performance, we carefully set the hyperparameters for each unlearning method across different datasets and architectures. For retrain-from-scratch, we followed the exact same training procedure as the original model, but only trained on the retain set, excluding the forget set. For Fine-tune, we trained the model for 10 epochs with a learning rate in the range [0.01, 0.1]. L1-sparse was run for 10 epochs with a learning rate in the range [0.005, 0.1] and a sparsity-promoting regularization parameter $\gamma$ in the range $[10^{-6}, 10^{-4}]$. NegGrad was also trained for 10 epochs, with a learning rate in the range $[10^{-4}, 0.1]$. For NegGrad+, we used a $\beta$ value in the range [0.85, 0.99] as a weighting factor that balances two components of the loss, training for 5 epochs with a learning rate in the range [0.01, 0.05]. Influence unlearning involved tuning the parameter $\alpha$ for the woodfisher Hessian Inverse approximation within the range [1, 100]. SalUn was trained for 5-10 epochs with a learning rate in the range [0.005, 0.1] and sparsity ratios in the range [0.3, 0.7]. Random-label was trained for 10 epochs with a learning rate in the range [0.01, 0.1].

In our experiments with forget / retain set partitions at varied levels of memorization or ES, we tuned the hyperparameters to achieve the best ToW performance for each unlearning algorithm. The results are reported as averages with 95% confidence intervals over 3 runs, except for relabeling-based methods, which had higher variance and were therefore run 6 times. For the RUM experiment, we adjusted the hyperparameters at each step to ensure that the accuracy after each step closely matched the accuracy obtained by retraining from scratch. This procedure was repeated for all algorithms, with results reported as averages over 3 runs with 95% confidence intervals.

### A.3 Procedure for creating retain / forget partitions with varying ES

To create retain / forget partitions with varied levels of ES, we followed a systematic procedure. Initially, we trained the original model $\theta^o$ on the entire training dataset $\mathcal{D}_{train}$. Using $\theta^o$, we then extracted embeddings for each data point in $\mathcal{D}_{train}$. The global centroid for $\mathcal{D}_{train}$, denoted as $\mu$, was determined by calculating the mean of all example embeddings. For each example $i$ in $\mathcal{D}_{train}$, we then computed its $l2$-distance from the global centroid $\mu$ in the original model's embedding space as follows:

$$d(i, \mu; \theta^o) = ||\phi_i - \mu||^2$$

We ranked these distances for all data examples in $\mathcal{D}_{train}$ and selected the 3000 examples with the highest distances to form the low ES bucket. Subsequently, we moved further down the ranked list, selecting 3000 examples with progressively lower distances to form the medium and high ES buckets, until we achieved the desired levels of ES variation. This approach allowed us to form forget sets, each with a size of 3000, categorized into low, medium, and high ES levels. The rationale behind this selection is that examples with high distances from the global centroid are considered "distant" from the overall data distribution in the embedding space and are therefore less entangled with the rest of the dataset, i.e., the retain set. Various unlearning algorithms were then deployed on $\theta^o$ across different forget / retain partitions. Their performance was measured using ToW along with forget, retain, and test accuracy, as well as MIA.

This procedure enabled us to create retain / forget partitions with varying ES values. The ES values for our low, medium, and high ES partitions are shown in Table 2. As observed from the table, the ES values increase from low to high ES partitions for both CIFAR-10 and CIFAR-100, confirming the effectiveness of our procedure. We also use Maximum Mean Discrepancy (MMD) [18], a widely-used metric, to further validate ES. The MMD is computed as follows: Given a pre-trained model $\theta^o$ (the original model in our case) trained on $\mathcal{D}_{train}$, we first extract features for the forget set $\mathcal{S}$ and the retain set $\mathcal{R}$ using $\theta^o$, denoted as $\phi(\mathcal{S})$ and $\phi(\mathcal{R})$, respectively. We then apply an RBF kernel to map these features to another space $\mathcal{H}$ such that $\phi'(\mathcal{S}), \phi'(\mathcal{R}) \in \mathcal{H}$. The MMD between $\phi(\mathcal{S})$ and $\phi(\mathcal{R})$ is then calculated using the following formula:

$$\text{MMD}^2(\phi(\mathcal{S}), \phi(\mathcal{R})) = \left\| \frac{1}{|\mathcal{S}|} \sum_{i \in \mathcal{S}} \phi'_i - \frac{1}{|\mathcal{R}|} \sum_{j \in \mathcal{R}} \phi'_j \right\|^2_{\mathcal{H}}$$

Our experiments reveal a negative correlation between ES and MMD scores, as reported in Table 2. For CIFAR-10/ResNet-18 and CIFAR-100/ResNet-50, low (high) ES values correspond to high (low) MMD values, supporting the consistency of the ES.

Additionally, Figure 6 presents the data representation of low, medium, and high ES partitions, confirming that the degree of entanglement between the retain and forget sets aligns with the computed ES values. As we move from low to high ES partitions, the forget set (yellow) and the retain set (blue) become increasingly entangled. This indicates that higher ES partitions reflect greater complexity in distinguishing between the two sets.

### A.4 Description of MIA and ToW-MIA

We adopted a MIA based on prediction confidence, following the procedure described by [26]. To conduct this attack, we first sampled equal-sized data from the retain set and the test set, using these to train a binary classifier as the MIA model. This model is designed to distinguish whether a data example was involved in the training stage or not.

Next, we applied this attack model to the forget set to evaluate unlearning performance during the testing phase, after an unlearning method was implemented. Intuitively, for successful unlearning, we want forget set examples to be classified as "non-training" data. We define "training" data as the positive class and "non-training" data as the negative class, and measured the performance of MIA by calculating the ratio of true negatives (i.e., the number of the forgetting samples predicted as

Table 2: ES and MMD scores for the low, medium and high forget / retain partitions for CIFAR-10 and CIFAR-100.

|  | Low ES | Medium ES | High ES |
| --- | --- | --- | --- |
| ES value | 309.94±98.56 | 1076.99±78.64 | 1612.210±110.82 |
| MMD value | $(5.15\pm0.26)\times10^{-2}$ | $(4.07\pm0.59)\times10^{-2}$ | $(2.84\pm0.75)\times10^{-2}$ |

(a) CIFAR-10

|  | Low ES | Medium ES | High ES |
| --- | --- | --- | --- |
| ES value | 963.82±113.53 | 2831.24±558.63 | 3876.90±426.92 |
| MMD value | $(1892.69\pm0.07)\times10^{-5}$ | $(1892.13\pm0.05)\times10^{-5}$ | $(1891.44\pm0.18)\times10^{-5}$ |

(b) CIFAR-100

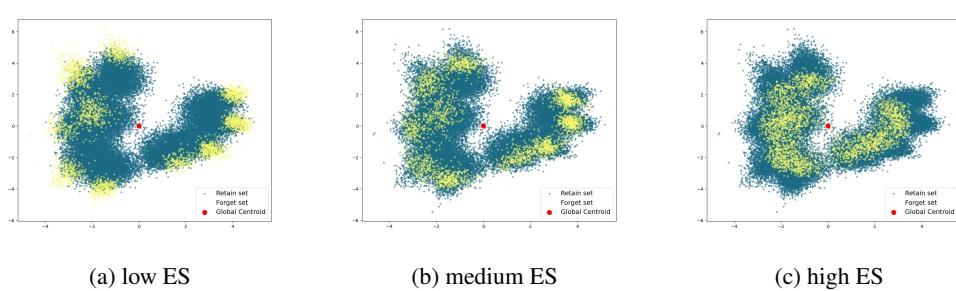

(a) low ES      (b) medium ES      (c) high ES

Figure 6: Data representation visualization for forget / retain partitions with low, medium, and high ES. We used PCA to reduce the dimensionality for visualization. In each figure, the data examples from the forget set are shown in yellow, while those from the retain set are in blue. The global centroid is marked in red at the center of the figures.

non-training examples) predicted by the MIA model to the total size of the forget set, as shown in (3), following the same procedure as prior work [26].

$$\text{MIA Performance} = \frac{TN_{\mathcal{S}}}{|\mathcal{S}|}, \tag{3}$$

In this context, $\mathcal{S} \subseteq \mathcal{D}_{train}$ represents the forget set and $|\mathcal{S}|$ is the total size of this set. The term $TN_{\mathcal{S}}$ denotes the number of true negatives predicted by the MIA model, indicating examples identified as "non-training" data. This metric ranges from 0 to 1, with higher values signifying a larger portion of the forget set predicted as "non-training" data. The ideal MIA score, for an unlearing algorithm, is one that matches the MIA score of retraining-from-scratch. Note that, even if applying this MIA on retrain-from-scratch, some portion of the forget set will be classified as "training data" due to heavily resembling the retain set (e.g. examples that are not really memorized may have similar confidences regardless on whether or not they were trained on). And we want the model confidences of the unlearned model to resemble as closely as possible those of retrain-from-scratch. For this reason, we additionally report the "MIA gap" (the absolute differentce between the MIA score for unlearning compared to that of retrain-from-scratch) as a easier-to-interpret metric, where lower is better, and the ideal score there is 0.

Building on the MIA setup above, we introduce another metric "ToW-MIA" to evaluate unlearning performance, similar to ToW described in Section 4. The metric is defined as follows:

$$\text{ToW-MIA}(\theta^u, \theta^r, \mathcal{S}, \mathcal{R}, \mathcal{D}_{test}) = (1 - \text{dm}(\theta^u, \theta^r, \mathcal{S})) \cdot (1 - \text{da}(\theta^u, \theta^r, \mathcal{R})) \cdot (1 - \text{da}(\theta^u, \theta^r, \mathcal{D}_{test}))$$

where $\text{m}(\theta, \mathcal{D})$ represents the MIA performance of the model $\theta$ trained on $\mathcal{D}$, as defined in Equation (3). The first term, $\text{dm}(\theta_u, \theta_r, \mathcal{S}) = |\text{m}(\theta_u, \mathcal{S}) - \text{m}(\theta_r, \mathcal{S})|$, denotes the absolute difference in MIA

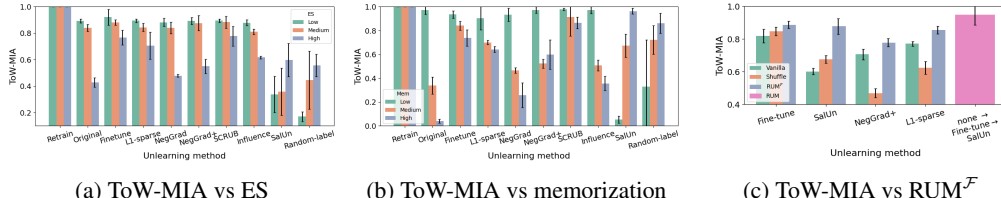

| (a) ToW-MIA vs ES | (b) ToW-MIA vs memorization | (c) ToW-MIA vs RUM$^{\mathcal{F}}$ |

Figure 7: Evaluating performance with ToW-MIA (higher is better) for unlearning difficulty analysis (a, b) and RUM$^{\mathcal{F}}$ (c). Error bars represent 95% confidence intervals from 3 runs per algorithm.

performance on $\mathcal{S}$ between the unlearned model $\theta_u$ and the retrained-from-scratch model $\theta_r$. This term is the key distinction between ToW and ToW-MIA, where "forget quality" is measured by MIA (ToW-MIA) rather than accuracy (ToW). The other two terms remain identical to those in ToW (see the equation in Section 4), representing the absolute difference in accuracy between $\theta^u$ and $\theta^r$ on $\mathcal{R}$ and $\mathcal{D}_{test}$, respectively, which reflect "model utility". Like ToW, ToW-MIA rewards unlearned models that approximate the performance of the retrained model across the forget, retain, and test sets. ToW-MIA ranges from 0 to 1, with higher values indicating better unlearning performance. Figure 7 show the same trends with ToW-MIA as for ToW, both for unlearning difficulty analysis and RUM. More detailed results using ToW-MIA can be found in Table 14.

## A.5  Data-space vs embedding-space outliers

Building on our discussion in Section 4, we identify two factors that affect the difficulty of unlearning: the entanglement between the forget and retain sets, and the memorization level of the forget examples. This raises the question: how do these two factors interact with each other? Are the outliers in data space the same as those in embedding space? Our findings in this section suggest that these factors are indeed distinct.

**Memorization within ES-based partitions**  We analyzed the memorization scores within each forget set categorized by different ES values. Figure 8 shows the distribution and average memorization scores in the forget sets of low, medium, and high ES categories. It can be seen that each ES category's forget set contains a mix of memorization levels. Specifically, while the mean memorization score of forget examples increases from low to high ES categories, the memorization scores still span the entire range from 0 to 1.

**Entanglement within memorization-based partitions**  We examined the ES values for each memorization category to understand the entanglement between the forget and retain sets in the embedding space. Table 3 presents the ES values corresponding to low, medium, and high memorization buckets. The findings reveal that all the memorization bucket exhibits relatively high ES, suggesting that separating outliers in the data space does not reduce the entanglement between the forget and retain sets in the embedding space.

Table 3: ES values for forget / retain partitions across varied memorization levels for CIFAR-10 and CIFAR-100.

|  | Low memorization | Medium memorization | High memorization |
|---|---|---|---|
| ES value | 21134.127 | 32785.711 | 14736.591 |

(a) CIFAR-10

|  | Low memorization | Medium memorization | High mamorization |
|---|---|---|---|
| ES value | 30028.924 | 58683.180 | 20528.561 |

(b) CIFAR-100

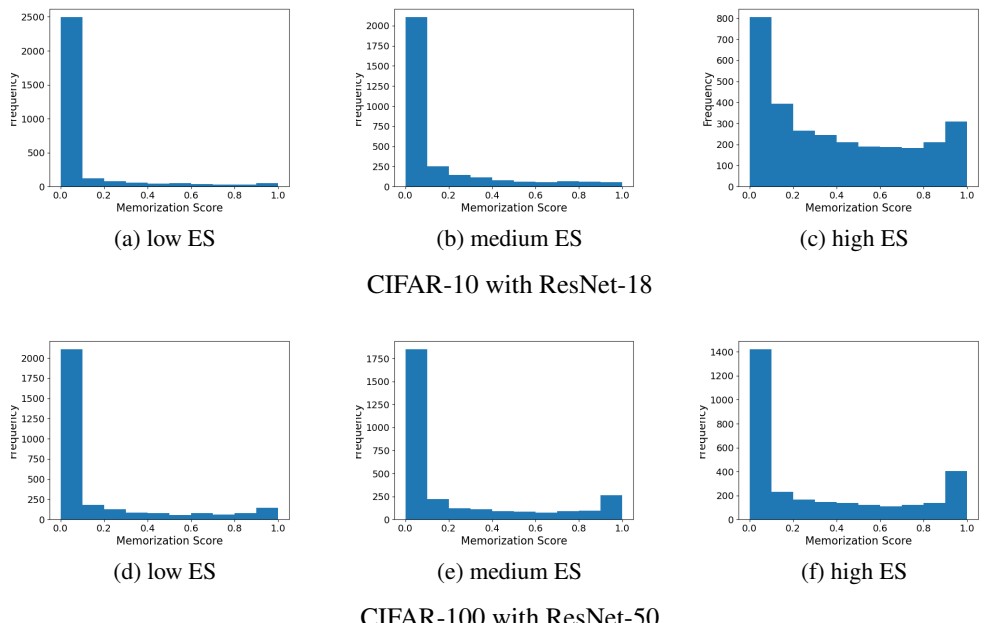

(a) low ES          (b) medium ES          (c) high ES

CIFAR-10 with ResNet-18

(d) low ES          (e) medium ES          (f) high ES

CIFAR-100 with ResNet-50

Figure 8: Memorization distribution for low, medium, high ES forget / retain partitions. The mean memorization score for low, medium, high ES partitions are $0.084\pm0.203$, $0.134\pm0.235$, $0.390\pm0.326$ for CIFAR-10, and $0.159\pm0.283$, $0.222\pm0.329$, and $0.317\pm0.364$ for CIFAR-100.

These findings demonstrate that embedding space entanglement and the memorization level of the forget set are distinct concepts, not merely different aspects of the same phenomenon.

### A.6 Confidence-based proxy for memorization

To improve the efficiency of calculating memorization scores, we leverage insights from previous works [23, 32] and introduce a model confidence-based metric, referred to as the **C-proxy**, which serves as a proxy for the memorization score. The computation of the C-proxy proceeds as follows: For each data point $(x_i, y_i) \in \mathcal{D}$, we track the softmax probability of the model's prediction $\theta(x_i)$ for the ground-truth label $y_i$ across all training epochs, as the model $\theta$ is trained on $\mathcal{D}$ using algorithm $\mathcal{A}$. These probabilities are then averaged over all epochs at the end of training to capture the training dynamics.

Table 4 presents the fidelity (measured by Spearman correlation) and efficiency (measured by additional computational cost) of the C-proxy in relation to memorization. This evaluation is conducted on both CIFAR-10 and CIFAR-100 datasets. As shown, the C-proxy demonstrates a strong negative Spearman correlation with memorization scores while significantly reducing computational overhead compared to both computing memorization scores and retraining the model from scratch. This suggests that the C-proxy is an effective proxy for memorization while also offering substantial efficiency gains.

Table 4: Comparison of C-proxy based on Spearman correlation with memorization and relative computation time percentages in comparison to memorization computation / retraining the model from scratch, evaluated on CIFAR-10 and CIFAR-100 using ResNet-18 and ResNet-50, respectively.

| Dataset | Spearman corr. (mem) | Comp. time % (mem) | Comp. time % (retrain) |
|---|---|---|---|
| CIFAR-10 | -0.80 | 0.018% | 17.123% |
| CIFAR-100 | -0.91 | 0.002% | 8.175% |

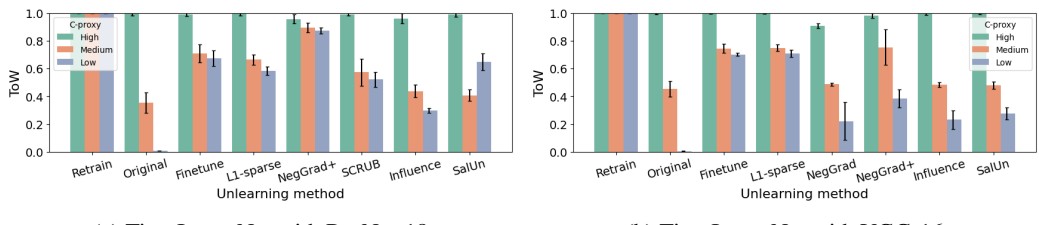

| (a) Tiny-ImageNet with ResNet-18 | (b) Tiny-ImageNet with VGG-16 |

Figure 9: Impact of C-proxy on unlearning difficulty: C-proxy vs ToW on Tiny-ImageNet across low, medium, and high C-proxy levels, using ResNet-18 and VGG-16. Error bars represent 95% confidence intervals from 3 runs per algorithm.

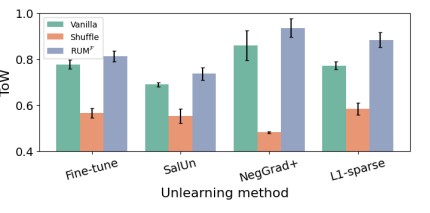 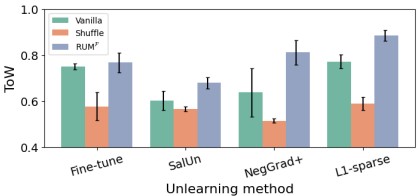

| (a) Tiny-ImageNet with ResNet-18 | (b) Tiny-ImageNet with VGG-16 |

Figure 10: Impact of C-proxy on unlearning performance on Tiny-ImageNet with ResNet-18 and VGG-16. Each unlearning algorithm is applied in three ways: $RUM^{\mathcal{F}}$, as well as vanilla and shuffle as comparison. Error bars represent 95% confidence intervals from 3 runs per algorithm.

Figure 4a and Figure 9 present results across different datasets and model architectures, demonstrating that the pattern identified in Section 4.2 remains consistent when using the C-proxy instead of the memorization score. Specifically, the lower the C-proxy value, the harder it is to unlearn (note that the C-proxy is negatively correlated with memorization). Furthermore, Figure 4b and Figure 10 show the results of $RUM^{\mathcal{F}}$ across various datasets and model architectures, using the C-proxy in place of memorization for the refinement step described in Section 5. More detailed results, including accuracy and MIA, are available in Table 17. These results demonstrate that $RUM^{\mathcal{F}}$ still achieves a significant unlearning performance gain when C-proxy is used, comparable to those obtained with the memorization score. A detailed analysis of various memorization proxies, their suitability, and their impact on RUM can be found in [37].

## A.7 Per-example differences between the unlearned and retrained models

While accuracy is a commonly used metric in the unlearning literature [29], it reflects an average over a data distribution and may overlook finer details. To gain a more granular understanding of the similarities/differences between the unlearned model and the retrained model, we examine their behavior at the level of individual examples. Specifically, we collect predictions from both models for each example and count the examples where their predictions differ. Table 5 presents these results for a representative setting: CIFAR-10 dataset with ResNet-18 architecture.

We observe that, in all cases, the percentage of disagreements between the unlearned and retrained models remains low. For highly memorized data (which typically makes unlearning harder), the disagreement rate increases but stays modest, ranging between ca. [7%, 15%]. For low-memorized data, the disagreement rate is lower, within ca. [3%, 5%].

Interestingly, for the vast majority of unlearning algorithms, these per-example disagreement trends align with those observed using ToW: low-memorized examples are generally easier to unlearn, as indicated by both ToW and the lower disagreement rate, while high-memorized examples are harder to unlearn, with higher disagreement rates. A similar trend is observed across different ES levels as well. This consistency between average accuracies (ToW) and per-example disagreements supports the robustness of our findings and underscores the validity of ToW as a measure of unlearning difficulty.

Table 5: ToW vs percentage of different predictions across ES or memorization levels on CIFAR-10/ResNet-18, averaged over 3 runs with 95% confidence intervals.

| | ToW | | | Percentage of different predictions (%) | | |
|---|---|---|---|---|---|---|
| | Low ES | Medium ES | High ES | Low ES | Medium ES | High ES |
| Original | $0.944 \pm 0.014$ | $0.928 \pm 0.022$ | $0.759 \pm 0.048$ | $0.329 \pm 0.069$ | $0.396 \pm 0.048$ | $1.514 \pm 0.212$ |
| Fine-tune | $0.952 \pm 0.024$ | $0.923 \pm 0.019$ | $0.908 \pm 0.018$ | $0.624 \pm 1.339$ | $1.676 \pm 2.431$ | $3.218 \pm 1.597$ |
| L1-sparse | $0.945 \pm 0.013$ | $0.926 \pm 0.019$ | $0.836 \pm 0.031$ | $0.335 \pm 0.051$ | $0.407 \pm 0.053$ | $6.367 \pm 11.164$ |
| NegGrad | $0.929 \pm 0.030$ | $0.887 \pm 0.047$ | $0.766 \pm 0.042$ | $0.479 \pm 0.413$ | $3.853 \pm 4.276$ | $3.594 \pm 3.802$ |
| NegGrad+ | $0.941 \pm 0.029$ | $0.920 \pm 0.038$ | $0.800 \pm 0.029$ | $0.370 \pm 0.022$ | $1.796 \pm 1.510$ | $1.752 \pm 0.174$ |
| Influence unlearning | $0.928 \pm 0.023$ | $0.890 \pm 0.012$ | $0.744 \pm 0.029$ | $0.362 \pm 0.208$ | $0.986 \pm 1.223$ | $3.381 \pm 3.342$ |
| SalUn | $0.783 \pm 0.031$ | $0.656 \pm 0.046$ | $0.618 \pm 0.056$ | $8.816 \pm 1.756$ | $1.752 \pm 5.706$ | $5.706 \pm 4.279$ |
| Random-label | $0.767 \pm 0.107$ | $0.663 \pm 0.055$ | $0.443 \pm 0.087$ | $5.566 \pm 0.406$ | $12.154 \pm 6.952$ | $0.579 \pm 0.579$ |

(a) Per-example difference vs ES

| | ToW | | | Percentage of different predictions (%) | | |
|---|---|---|---|---|---|---|
| | Low mem | Medium mem | High mem | Low mem | Medium mem | High mem |
| Original | $0.988 \pm 0.007$ | $0.723 \pm 0.053$ | $0.231 \pm 0.058$ | $3.611 \pm 0.856$ | $5.081 \pm 0.550$ | $7.341 \pm 0.822$ |
| Fine-tune | $0.933 \pm 0.052$ | $0.884 \pm 0.019$ | $0.760 \pm 0.065$ | $5.962 \pm 1.525$ | $6.359 \pm 1.135$ | $7.099 \pm 2.370$ |
| L1-sparse | $0.914 \pm 0.061$ | $0.816 \pm 0.011$ | $0.629 \pm 0.087$ | $6.668 \pm 1.676$ | $10.163 \pm 3.705$ | $9.834 \pm 2.030$ |
| NegGrad | $0.938 \pm 0.028$ | $0.738 \pm 0.005$ | $0.325 \pm 0.098$ | $5.412 \pm 1.020$ | $9.073 \pm 1.283$ | $14.256 \pm 3.437$ |
| NegGrad+ | $0.965 \pm 0.020$ | $0.831 \pm 0.032$ | $0.661 \pm 0.082$ | $4.288 \pm 0.452$ | $6.357 \pm 1.023$ | $12.190 \pm 2.720$ |
| Influence unlearning | $0.986 \pm 0.031$ | $0.738 \pm 0.051$ | $0.381 \pm 0.037$ | $3.964 \pm 0.803$ | $9.668 \pm 2.469$ | $18.484 \pm 1.491$ |
| SalUn | $0.774 \pm 0.043$ | $0.758 \pm 0.053$ | $0.886 \pm 0.082$ | $5.963 \pm 1.281$ | $7.507 \pm 0.268$ | $8.031 \pm 0.540$ |
| Random-label | $0.709 \pm 0.029$ | $0.548 \pm 0.093$ | $0.877 \pm 0.064$ | $13.491 \pm 5.363$ | $12.973 \pm 1.635$ | $12.905 \pm 4.413$ |

(b) Per-example difference vs memorization

## A.8   Detailed results

In this section, we show the results that were used to construct all figures in the paper, as well as additional results and analyses.

**Entanglement of forget and retain sets affects unlearning difficulty**   The first factor affecting unlearning difficulty, as discussed in Section 4.1, is the degree of entanglement between the forget and retain sets in the embedding space. Specifically, the more entangled these sets are, the harder it becomes to unlearn. We present detailed results on how this factor affects unlearning difficulty in Table 6, 7, 8, 9, 10 and Figure 11. Table 6 displays the ToW results for different ES partitions across various settings, including CIFAR-10, CIFAR-100 and Tiny-ImageNet datasets, as well as ResNet and VGG architectures. Additionally, comprehensive data—covering forget, retain, and test accuracy along with MIA performance—are provided in Figure 11 and in Table 7 through Table 10 for these datasets and model architectures. All the results are averaged over 3 runs for each algorithm (6 runs for relabelling-based algorithms due to their higher variance), along with 95% confidence intervals.

**Memorization of forget examples affects unlearning difficulty**   In Section 4.2, we discussed another critical factor affecting unlearning difficulty: the memorization level of the forget examples. Specifically, when a forget set consists of examples that are more highly memorized by the model, it becomes more difficult to unlearn (for most algorithms). The detailed results that support this observation are presented in Table 11, Table 12, Table 13, and Figure 12.

Table 11 illustrates the ToW for forget sets with varying memorization levels. Additionally, Figure 12 and Table 12 to Table 13 provide extensive data on forget, retain, and test accuracy, as well as MIA performance. For all the experiments, the average results are reported with 95% confidence intervals over 6 runs for relabelling-based algorithms and 3 runs for others.

**RUM experiments**   We present the details of our RUM experiment in this section. Table 15 presents the distribution of examples in the forget set for each class in the RUM experiments for both CIFAR-10 and CIFAR-100 datasets. The size of the forget set is 3000 in all the experiments. For CIFAR-10, the table lists the number of forget set examples for each class. For CIFAR-100, the table arranges the number of forget set examples in a 20x5 format for better readability. As shown in Table 15, the forget set covers examples from all classes in both CIFAR-10 and CIFAR-100 experiments.

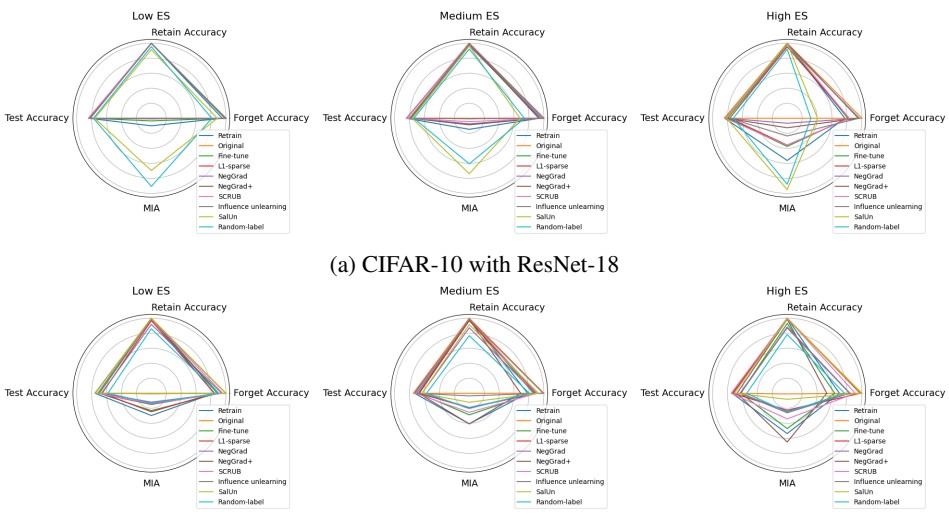

(a) CIFAR-10 with ResNet-18

(b) CIFAR-100 with ResNet-50

Figure 11: Forget, retain, test accuracy and MIA performance for forget / retain partitions with varied ES, for CIFAR-10 using ResNet-18 and CIFAR-100 using ResNet-50. As the ES increases, there is a notable decline in forget accuracy and a significant rise in MIA performance. This trend indicates that as the forget and retain sets become more entangled, more information from the forget set is effectively removed after unlearning.

Tables 16 provide detailed results when applying RUM to different unlearning algorithms for CIFAR-10 using ResNet-18 and CIFAR-100 using ResNet-50. We selected the top-performing algorithms from previous experiments (i.e., Fine-tune, L1-sparse, NegGrad+, SalUn) and applied the refinement strategy to each algorithm in three ways: i) "vanilla": unlearning $\mathcal{S}$ in one go, ii) "shuffle": sequentially applying the algorithm on 3 equal-sized random subsets, serving as a control experiment, iii) "RUM$^{\mathcal{F}}$": sequentially applying the algorithm on 3 subsets of $\mathcal{S}$ in low $\to$ med $\to$ high memorization order. Each experiment was conducted 3 times, with average values and 95% confidence intervals reported. Our results indicate that RUM$^{\mathcal{F}}$ improves the performance of each unlearning algorithm, and the full RUM approach, which uses the best algorithm for each subset, achieves the overall best results.

Furthermore, to gain a deeper understanding of the dynamics involved in applying RUM, we plotted the sequence dynamics for SalUn RUM$^{\mathcal{F}}$ on CIFAR-10 (Figure 5) and NegGrad+ RUM$^{\mathcal{F}}$ on CIFAR-100 (Figure 13). These plots show the accuracy of the entire forget set, the retain set, the test set, and subsets of the forget set after each step. They demonstrate that while both orderings (low $\to$ med $\to$ high and high $\to$ med $\to$ low memorization) yield similar ToW according to Table 16, their sequence dynamics during the unlearning phase are different. This phenomenon is discussed in Section 6 in the main paper, specifically in the **Analysis of Sequence Dynamics** paragraph.

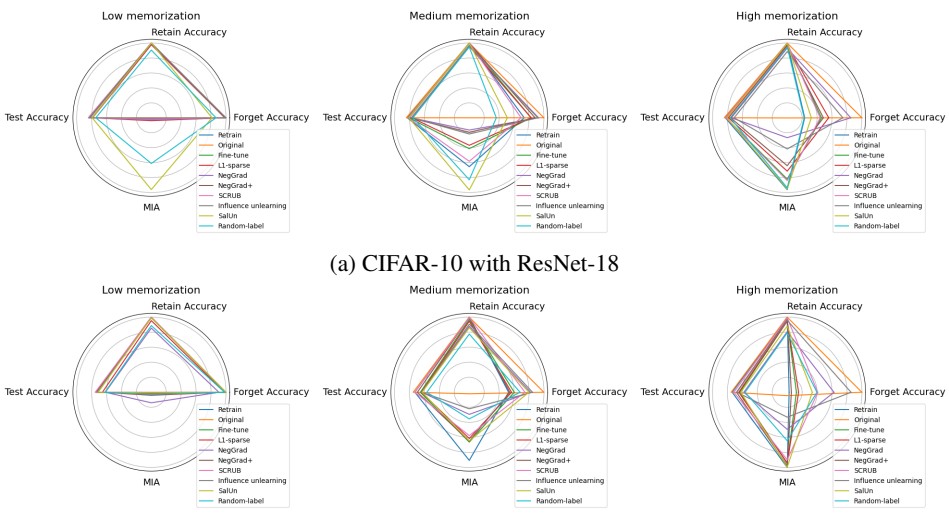

(a) CIFAR-10 with ResNet-18

(b) CIFAR-100 with ResNet-50

Figure 12: Forget, retain, test accuracy and MIA performance for forget / retain partitions with varying levels of memorization, for CIFAR-10 using ResNet-18 and CIFAR-100 using ResNet-50. As the memorization level of the forget sets increases, forget accuracy significantly decreases while MIA performance increases. This trend indicates that as the forget examples become more memorized by the model, unlearning becomes more effective in removing the effect of these examples.

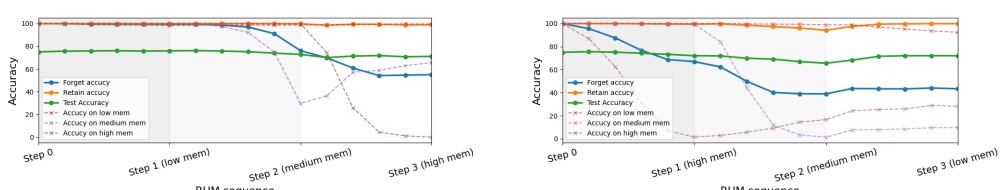

(a) low → med → high (Tow: 0.92, MIA gap: 0.059)  (b) high → med → low (ToW: 0.93, MIA gap: 0.022)

Figure 13: Analysis of sequence dynamics for two different orderings. We apply NegGrad+ RUM on CIFAR-100 dataset using ResNet-50 and show the accuracy on overall forget set (and each of its subsets), the retain set and test set after each step in the RUM sequence.

Table 6: ToW for different unlearning algorithms applied to forget / retain sets with varying ES, evaluated on various datasets and model architectures. Across all algorithms, including the baseline without any unlearning performed (denoted as "Original"), we observe that ToW decreases from low to high ES partitions, indicating that unlearning becomes harder as the forget and retain sets become more entangled.

|  | Low ES | Medium ES | High ES |
|---|---|---|---|
| Retrain | $1.000 \pm 0.000$ | $1.000 \pm 0.000$ | $1.000 \pm 0.000$ |
| Original | $0.944 \pm 0.014$ | $0.928 \pm 0.022$ | $0.759 \pm 0.048$ |
| Fine-tune | $0.952 \pm 0.024$ | $0.923 \pm 0.019$ | $0.908 \pm 0.018$ |
| L1-sparse | $0.945 \pm 0.013$ | $0.926 \pm 0.019$ | $0.836 \pm 0.031$ |
| NegGrad | $0.929 \pm 0.030$ | $0.887 \pm 0.047$ | $0.766 \pm 0.042$ |
| NegGrad+ | $0.941 \pm 0.029$ | $0.920 \pm 0.038$ | $0.800 \pm 0.029$ |
| SCRUB | $0.944 \pm 0.014$ | $0.939 \pm 0.021$ | $0.920 \pm 0.033$ |
| Influence Unlearning | $0.928 \pm 0.023$ | $0.890 \pm 0.012$ | $0.744 \pm 0.029$ |
| SalUn | $0.783 \pm 0.031$ | $0.656 \pm 0.046$ | $0.618 \pm 0.056$ |
| Random-label | $0.767 \pm 0.107$ | $0.663 \pm 0.055$ | $0.443 \pm 0.087$ |

(a) CIFAR-10 with ResNet-18

|  | Low ES | Medium ES | High ES |
|---|---|---|---|
| Retrain | $1.000 \pm 0.000$ | $1.000 \pm 0.000$ | $1.000 \pm 0.000$ |
| Original | $0.836 \pm 0.043$ | $0.768 \pm 0.036$ | $0.690 \pm 0.061$ |
| Fine-tune | $0.878 \pm 0.014$ | $0.852 \pm 0.006$ | $0.773 \pm 0.029$ |
| L1-sparse | $0.867 \pm 0.039$ | $0.803 \pm 0.025$ | $0.709 \pm 0.057$ |
| NegGrad | $0.755 \pm 0.027$ | $0.707 \pm 0.039$ | $0.666 \pm 0.060$ |
| NegGrad+ | $0.922 \pm 0.055$ | $0.844 \pm 0.014$ | $0.766 \pm 0.022$ |
| SCRUB | $0.842 \pm 0.089$ | $0.831 \pm 0.068$ | $0.781 \pm 0.057$ |
| Influence Unlearning | $0.806 \pm 0.027$ | $0.756 \pm 0.041$ | $0.668 \pm 0.037$ |
| SalUn | $0.835 \pm 0.037$ | $0.704 \pm 0.028$ | $0.678 \pm 0.042$ |
| Random-label | $0.675 \pm 0.029$ | $0.629 \pm 0.031$ | $0.585 \pm 0.014$ |

(b) CIFAR-100 with ResNet-50

|  | Low ES | Medium ES | High ES |
|---|---|---|---|
| Retrain | $1.000 \pm 0.000$ | $1.000 \pm 0.000$ | $1.000 \pm 0.000$ |
| Original | $0.920 \pm 0.019$ | $0.626 \pm 0.046$ | $0.377 \pm 0.015$ |
| Fine-tune | $0.919 \pm 0.015$ | $0.770 \pm 0.039$ | $0.765 \pm 0.022$ |
| L1-sparse | $0.920 \pm 0.018$ | $0.884 \pm 0.014$ | $0.866 \pm 0.004$ |
| NegGrad | $0.895 \pm 0.032$ | $0.616 \pm 0.070$ | $0.476 \pm 0.027$ |
| NegGrad+ | $0.921 \pm 0.021$ | $0.914 \pm 0.041$ | $0.854 \pm 0.021$ |
| Influence Unlearning | $0.892 \pm 0.029$ | $0.595 \pm 0.019$ | $0.442 \pm 0.026$ |
| SalUn | $0.920 \pm 0.018$ | $0.613 \pm 0.044$ | $0.579 \pm 0.021$ |

(c) Tiny-ImageNet with ResNet-18

|  | Low ES | Medium ES | High ES |
|---|---|---|---|
| Retrain | $1.000 \pm 0.000$ | $1.000 \pm 0.000$ | $1.000 \pm 0.000$ |
| Original | $0.933 \pm 0.002$ | $0.568 \pm 0.017$ | $0.331 \pm 0.030$ |
| Fine-tune | $0.936 \pm 0.003$ | $0.749 \pm 0.055$ | $0.677 \pm 0.087$ |
| L1-sparse | $0.936 \pm 0.006$ | $0.763 \pm 0.043$ | $0.737 \pm 0.026$ |
| NegGrad | $0.919 \pm 0.019$ | $0.569 \pm 0.037$ | $0.379 \pm 0.016$ |
| NegGrad+ | $0.934 \pm 0.015$ | $0.806 \pm 0.027$ | $0.577 \pm 0.181$ |
| Influence Unlearning | $0.933 \pm 0.005$ | $0.547 \pm 0.041$ | $0.423 \pm 0.046$ |
| SalUn | $0.934 \pm 0.001$ | $0.554 \pm 0.010$ | $0.411 \pm 0.118$ |

(d) Tiny-ImageNet with VGG-16

Table 7: Accuracy and MIA performance for different unlearning algorithms applied to forget / retain sets with varying ES for CIFAR-10 using ResNet-18.

|  | Forget Acc | Retain Acc | Test Acc | MIA |
|---|---|---|---|---|
| Retrain | $95.078 \pm 0.995$ | $100.000 \pm 0.003$ | $84.040 \pm 1.387$ | $0.100 \pm 0.010$ |
| Original | $100.000 \pm 0.000$ | $100.000 \pm 0.000$ | $84.353 \pm 3.455$ | $0.000 \pm 0.000$ |
| Fine-tune | $98.522 \pm 3.296$ | $99.675 \pm 1.385$ | $83.107 \pm 3.862$ | $0.035 \pm 0.074$ |
| L1-sparse | $100.000 \pm 0.000$ | $99.994 \pm 0.022$ | $83.787 \pm 3.632$ | $0.001 \pm 0.001$ |
| NegGrad | $99.800 \pm 0.461$ | $99.853 \pm 0.429$ | $81.740 \pm 5.270$ | $0.006 \pm 0.007$ |
| NegGrad+ | $99.500 \pm 0.647$ | $99.988 \pm 0.051$ | $82.743 \pm 6.193$ | $0.007 \pm 0.007$ |
| SCRUB | $99.978 \pm 0.096$ | $100.000 \pm 0.000$ | $84.280 \pm 3.505$ | $0.002 \pm 0.001$ |
| Influence unlearning | $100.000 \pm 0.000$ | $99.964 \pm 0.154$ | $81.667 \pm 2.547$ | $0.000 \pm 0.001$ |
| SalUn | $88.400 \pm 6.843$ | $91.442 \pm 2.438$ | $75.833 \pm 0.725$ | $0.696 \pm 0.172$ |
| Random-label | $81.100 \pm 10.427$ | $95.396 \pm 0.682$ | $77.470 \pm 4.529$ | $0.908 \pm 0.026$ |

(a) Low ES

|  | Forget Acc | Retain Acc | Test Acc | MIA |
|---|---|---|---|---|
| Retrain | $94.067 \pm 0.722$ | $100.000 \pm 0.000$ | $84.167 \pm 1.243$ | $0.147 \pm 0.009$ |
| Original | $100.000 \pm 0.000$ | $100.000 \pm 0.000$ | $84.353 \pm 3.455$ | $0.000 \pm 0.000$ |
| Fine-tune | $96.678 \pm 3.525$ | $98.655 \pm 2.575$ | $80.210 \pm 3.004$ | $0.078 \pm 0.019$ |
| L1-sparse | $99.978 \pm 0.096$ | $99.990 \pm 0.044$ | $83.580 \pm 3.270$ | $0.005 \pm 0.014$ |
| NegGrad | $96.300 \pm 4.774$ | $96.527 \pm 4.294$ | $78.163 \pm 4.326$ | $0.074 \pm 0.057$ |
| NegGrad+ | $93.200 \pm 4.241$ | $98.907 \pm 1.839$ | $78.580 \pm 4.404$ | $0.087 \pm 0.041$ |
| SCRUB | $98.756 \pm 1.273$ | $100.000 \pm 0.000$ | $84.180 \pm 4.083$ | $0.045 \pm 0.056$ |
| Influence unlearning | $99.889 \pm 0.172$ | $99.372 \pm 0.774$ | $79.260 \pm 0.774$ | $0.007 \pm 0.005$ |
| SalUn | $68.044 \pm 7.008$ | $96.222 \pm 6.649$ | $76.380 \pm 6.843$ | $0.739 \pm 0.270$ |
| Random-label | $73.922 \pm 3.527$ | $92.244 \pm 7.643$ | $74.157 \pm 10.427$ | $0.607 \pm 0.316$ |

(b) Medium ES

|  | Forget Acc | Retain Acc | Test Acc | MIA |
|---|---|---|---|---|
| Retrain | $77.300 \pm 3.130$ | $100.000 \pm 0.003$ | $83.653 \pm 2.280$ | $0.562 \pm 0.026$ |
| Original | $100.000 \pm 0.000$ | $100.000 \pm 0.000$ | $84.353 \pm 3.455$ | $0.000 \pm 0.001$ |
| Fine-tune | $81.289 \pm 2.298$ | $98.371 \pm 1.395$ | $79.763 \pm 1.587$ | $0.375 \pm 0.066$ |
| L1-sparse | $83.022 \pm 14.615$ | $95.164 \pm 10.943$ | $76.943 \pm 6.159$ | $0.357 \pm 0.075$ |
| NegGrad | $96.756 \pm 3.705$ | $97.925 \pm 3.977$ | $80.927 \pm 6.576$ | $0.066 \pm 0.052$ |
| NegGrad+ | $95.222 \pm 2.993$ | $99.966 \pm 0.065$ | $81.447 \pm 3.720$ | $0.126 \pm 0.055$ |
| SCRUB | $82.967 \pm 3.929$ | $99.920 \pm 0.330$ | $82.717 \pm 7.032$ | $0.360 \pm 0.109$ |
| Influence unlearning | $96.111 \pm 6.259$ | $98.161 \pm 3.205$ | $77.093 \pm 5.087$ | $0.236 \pm 0.044$ |
| SalUn | $40.722 \pm 11.043$ | $99.967 \pm 0.080$ | $81.250 \pm 3.335$ | $0.951 \pm 0.077$ |
| Random-label | $31.656 \pm 5.551$ | $91.871 \pm 0.275$ | $72.273 \pm 1.895$ | $0.879 \pm 0.055$ |

(c) High ES

Table 8: Accuracy and MIA performance of different unlearning algorithms on forget / retain sets with varying ES for CIFAR-100 using ResNet-50.

| | Forget Acc | Retain Acc | Test Acc | MIA |
|---|---|---|---|---|
| Retrain | 84.222 ± 3.734 | 99.968 ± 0.012 | 75.493 ± 0.756 | 0.296 ± 0.041 |
| Original | 100.000 ± 0.000 | 99.959 ± 0.015 | 75.003 ± 2.809 | 0.000 ± 0.000 |
| Fine-tune | 90.044 ± 8.075 | 99.148 ± 1.639 | 69.577 ± 5.366 | 0.231 ± 0.067 |
| L1-sparse | 84.611 ± 6.139 | 96.807 ± 2.881 | 65.840 ± 3.228 | 0.246 ± 0.040 |
| NegGrad | 94.200 ± 2.319 | 91.843 ± 1.718 | 66.793 ± 3.820 | 0.118 ± 0.022 |
| NegGrad+ | 88.111 ± 1.532 | 99.841 ± 0.162 | 71.493 ± 2.158 | 0.149 ± 0.028 |
| SCRUB | 95.556 ± 6.560 | 98.962 ± 4.251 | 71.510 ± 12.504 | 0.137 ± 0.059 |
| Influence unlearning | 99.722 ± 0.981 | 99.202 ± 2.531 | 71.610 ± 5.558 | 0.007 ± 0.020 |
| SalUn | 100.000 ± 0.000 | 99.952 ± 0.016 | 74.707 ± 2.064 | 0.006 ± 0.002 |
| Random-label | 89.267 ± 1.159 | 85.992 ± 0.557 | 58.190 ± 1.640 | 0.130 ± 0.023 |

(a) Low ES

| | Forget Acc | Retain Acc | Test Acc | MIA |
|---|---|---|---|---|
| Retrain | 78.133 ± 1.616 | 99.960 ± 0.009 | 73.273 ± 2.131 | 0.407 ± 0.007 |
| Original | 100.000 ± 0.000 | 99.959 ± 0.015 | 75.003 ± 2.809 | 0.001 ± 0.001 |
| Fine-tune | 86.078 ± 8.732 | 98.069 ± 4.782 | 67.680 ± 4.230 | 0.287 ± 0.040 |
| L1-sparse | 89.511 ± 4.761 | 96.656 ± 2.561 | 66.983 ± 3.893 | 0.195 ± 0.025 |
| NegGrad | 88.289 ± 1.800 | 87.242 ± 2.187 | 63.380 ± 3.313 | 0.189 ± 0.018 |
| NegGrad+ | 68.900 ± 2.582 | 98.251 ± 1.412 | 67.913 ± 1.354 | 0.408 ± 0.035 |
| SCRUB | 90.656 ± 3.795 | 98.950 ± 4.390 | 71.863 ± 14.570 | 0.254 ± 0.100 |
| Influence unlearning | 98.978 ± 1.962 | 98.613 ± 2.268 | 70.093 ± 0.981 | 0.035 ± 0.042 |
| SalUn | 92.744 ± 1.706 | 91.400 ± 1.657 | 63.460 ± 1.319 | 0.120 ± 0.011 |
| Random-label | 80.622 ± 0.751 | 76.779 ± 0.664 | 57.213 ± 1.410 | 0.204 ± 0.004 |

(b) Medium ES

| | Forget Acc | Retain Acc | Test Acc | MIA |
|---|---|---|---|---|
| Retrain | 69.789 ± 6.038 | 99.963 ± 0.012 | 73.827 ± 3.116 | 0.536 ± 0.073 |
| Original | 99.989 ± 0.048 | 99.960 ± 0.018 | 75.003 ± 2.809 | 0.009 ± 0.007 |
| Fine-tune | 63.622 ± 6.326 | 93.054 ± 2.972 | 62.320 ± 3.726 | 0.470 ± 0.060 |
| L1-sparse | 97.856 ± 1.519 | 99.706 ± 0.286 | 72.620 ± 1.985 | 0.225 ± 0.057 |
| NegGrad | 83.489 ± 23.803 | 87.013 ± 19.000 | 63.343 ± 12.854 | 0.260 ± 0.245 |
| NegGrad+ | 52.944 ± 5.967 | 98.491 ± 0.732 | 67.257 ± 2.996 | 0.649 ± 0.091 |
| SCRUB | 91.000 ± 0.865 | 99.962 ± 0.010 | 74.703 ± 3.247 | 0.341 ± 0.038 |
| Influence unlearning | 84.133 ± 8.032 | 88.398 ± 5.947 | 62.170 ± 1.267 | 0.242 ± 0.046 |
| SalUn | 99.167 ± 1.444 | 99.289 ± 1.222 | 70.567 ± 2.274 | 0.080 ± 0.058 |
| Random-label | 76.756 ± 4.633 | 78.256 ± 4.378 | 54.160 ± 1.832 | 0.250 ± 0.041 |

(c) High ES

Table 9: Accuracy and MIA performance of different unlearning algorithms on forget / retain sets with varying ES for Tiny-ImageNet using ResNet-18.

|  | Forget Acc | Retain Acc | Test Acc | MIA |
|---|---|---|---|---|
| Retrain | 92.175 ± 1.804 | 99.979 ± 0.001 | 64.593 ± 1.146 | 0.164 ± 0.016 |
| Original | 100.000 ± 0.000 | 99.979 ± 0.000 | 64.773 ± 1.027 | 0.001 ± 0.001 |
| Fine-tune | 100.000 ± 0.000 | 99.979 ± 0.000 | 64.253 ± 0.553 | 0.007 ± 0.003 |
| L1-sparse | 100.000 ± 0.000 | 99.979 ± 0.000 | 64.393 ± 0.596 | 0.001 ± 0.002 |
| NegGrad | 99.600 ± 0.166 | 98.858 ± 0.319 | 62.386 ± 1.957 | 0.011 ± 0.004 |
| NegGrad+ | 100.000 ± 0.000 | 99.979 ± 0.000 | 64.506 ± 1.275 | 0.000 ± 0.000 |
| Influence unlearning | 99.789 ± 0.048 | 99.637 ± 0.492 | 61.452 ± 2.375 | 0.009 ± 0.006 |
| SalUn | 100.000 ± 0.000 | 99.979 ± 0.000 | 64.513 ± 1.446 | 0.000 ± 0.000 |

(a) Low ES

|  | Forget Acc | Retain Acc | Test Acc | MIA |
|---|---|---|---|---|
| Retrain | 62.800 ± 4.303 | 99.980 ± 0.003 | 64.480 ± 0.698 | 0.670 ± 0.031 |
| Original | 99.978 ± 0.048 | 99.980 ± 0.001 | 64.773 ± 1.027 | 0.010 ± 0.009 |
| Fine-tune | 85.256 ± 2.155 | 99.980 ± 0.001 | 63.893 ± 1.599 | 0.656 ± 0.010 |
| L1-sparse | 64.989 ± 7.874 | 97.985 ± 1.636 | 56.718 ± 2.988 | 0.502 ± 0.015 |
| NegGrad | 89.789 ± 9.586 | 90.936 ± 10.713 | 57.258 ± 7.332 | 0.165 ± 0.074 |
| NegGrad+ | 68.878 ± 2.883 | 99.977 ± 0.007 | 61.792 ± 0.516 | 0.495 ± 0.049 |
| Influence unlearning | 82.744 ± 2.567 | 86.616 ± 3.098 | 50.297 ± 3.316 | 0.145 ± 0.102 |
| SalUn | 87.344 ± 3.792 | 86.758 ± 4.405 | 58.045 ± 0.977 | 0.689 ± 0.166 |

(b) Medium ES

|  | Forget Acc | Retain Acc | Test Acc | MIA |
|---|---|---|---|---|
| Retrain | 37.822 ± 1.112 | 99.986 ± 0.006 | 64.313 ± 0.553 | 0.891 ± 0.006 |
| Original | 99.889 ± 0.172 | 99.983 ± 0.005 | 64.773 ± 1.027 | 0.061 ± 0.041 |
| Fine-tune | 60.889 ± 1.574 | 99.983 ± 0.003 | 63.799 ± 1.499 | 0.934 ± 0.008 |
| L1-sparse | 45.811 ± 0.912 | 99.260 ± 0.538 | 59.132 ± 0.849 | 0.813 ± 0.076 |
| NegGrad | 67.989 ± 8.858 | 80.444 ± 8.129 | 49.230 ± 4.824 | 0.457 ± 0.101 |
| NegGrad+ | 49.933 ± 2.191 | 99.982 ± 0.006 | 61.432 ± 0.928 | 0.843 ± 0.025 |
| Influence unlearning | 78.678 ± 13.145 | 86.881 ± 11.277 | 50.703 ± 5.029 | 0.640 ± 0.091 |
| SalUn | 62.000 ± 6.416 | 82.350 ± 2.534 | 57.091 ± 2.965 | 0.926 ± 0.056 |

(c) High ES

Table 10: Accuracy and MIA performance of different unlearning algorithms on forget / retain sets with varying ES for Tiny-ImageNet using VGG-16.

| | Forget Acc | Retain Acc | Test Acc | MIA |
|---|---|---|---|---|
| Retrain | 93.942 ± 0.752 | 99.979 ± 0.001 | 60.279 ± 1.269 | 0.119 ± 0.017 |
| Original | 100.000 ± 0.000 | 99.979 ± 0.001 | 60.479 ± 0.938 | 0.001 ± 0.003 |
| Fine-tune | 100.000 ± 0.000 | 99.979 ± 0.000 | 60.332 ± 0.398 | 0.001 ± 0.001 |
| L1-sparse | 100.000 ± 0.000 | 99.979 ± 0.000 | 60.225 ± 0.331 | 0.001 ± 0.000 |
| NegGrad | 99.978 ± 0.048 | 99.653 ± 0.216 | 58.445 ± 0.451 | 0.003 ± 0.003 |
| NegGrad+ | 100.000 ± 0.000 | 99.979 ± 0.001 | 59.899 ± 0.748 | 0.001 ± 0.002 |
| Influence unlearning | 99.989 ± 0.048 | 99.979 ± 0.001 | 60.412 ± 1.172 | 0.001 ± 0.001 |
| SalUn | 100.000 ± 0.000 | 99.979 ± 0.000 | 60.372 ± 1.093 | 0.001 ± 0.000 |

(a) Low ES

| | Forget Acc | Retain Acc | Test Acc | MIA |
|---|---|---|---|---|
| Retrain | 56.978 ± 2.110 | 99.980 ± 0.003 | 60.139 ± 0.445 | 0.662 ± 0.043 |
| Original | 99.989 ± 0.048 | 99.979 ± 0.000 | 60.479 ± 0.938 | 0.008 ± 0.001 |
| Fine-tune | 57.133 ± 5.043 | 87.325 ± 4.327 | 47.283 ± 1.898 | 0.464 ± 0.033 |
| L1-sparse | 61.633 ± 3.340 | 89.064 ± 0.687 | 49.983 ± 0.875 | 0.436 ± 0.049 |
| NegGrad | 88.200 ± 20.298 | 90.202 ± 20.598 | 52.604 ± 12.745 | 0.167 ± 0.189 |
| NegGrad+ | 69.944 ± 4.170 | 99.367 ± 0.642 | 53.284 ± 1.557 | 0.413 ± 0.066 |
| Influence unlearning | 81.922 ± 25.731 | 84.474 ± 24.099 | 47.469 ± 11.033 | 0.178 ± 0.081 |
| SalUn | 78.122 ± 6.099 | 78.968 ± 4.343 | 49.116 ± 0.508 | 0.251 ± 0.089 |

(b) Medium ES

| | Forget Acc | Retain Acc | Test Acc | MIA |
|---|---|---|---|---|
| Retrain | 33.289 ± 2.942 | 99.982 ± 0.004 | 59.859 ± 0.959 | 0.880 ± 0.039 |
| Original | 99.933 ± 0.083 | 99.981 ± 0.001 | 60.479 ± 0.938 | 0.034 ± 0.010 |
| Fine-tune | 30.089 ± 5.122 | 81.706 ± 5.794 | 45.336 ± 2.339 | 0.709 ± 0.055 |
| L1-sparse | 40.300 ± 3.354 | 88.805 ± 2.483 | 49.130 ± 0.358 | 0.688 ± 0.053 |
| NegGrad | 93.256 ± 7.453 | 97.402 ± 4.942 | 57.151 ± 2.957 | 0.169 ± 0.124 |
| NegGrad+ | 73.489 ± 23.035 | 99.802 ± 0.736 | 56.711 ± 3.155 | 0.513 ± 0.406 |
| Influence unlearning | 82.933 ± 4.683 | 92.497 ± 1.794 | 50.690 ± 0.650 | 0.233 ± 0.249 |
| SalUn | 84.089 ± 23.491 | 91.948 ± 13.228 | 51.810 ± 9.754 | 0.334 ± 0.193 |

(c) High ES

Table 11: ToW for different unlearning algorithms applied to forget sets with varying levels of memorization, for CIFAR-10 using ResNet-18 and CIFAR-100 using ResNet-50. As the memorization level of the forget examples increases, the ToW significantly decreases for most algorithms, indicating that unlearning becomes harder when the forget examples are more memorized by the model.

|  | Low memorization | Medium memorization | High memorization |
|---|---|---|---|
| Retrain | $1.000 \pm 0.000$ | $1.000 \pm 0.000$ | $1.000 \pm 0.000$ |
| Original | $0.988 \pm 0.007$ | $0.723 \pm 0.053$ | $0.231 \pm 0.058$ |
| Fine-tune | $0.933 \pm 0.052$ | $0.884 \pm 0.019$ | $0.760 \pm 0.065$ |
| L1-sparse | $0.914 \pm 0.061$ | $0.816 \pm 0.011$ | $0.629 \pm 0.087$ |
| NegGrad | $0.938 \pm 0.028$ | $0.738 \pm 0.005$ | $0.325 \pm 0.098$ |
| NegGrad+ | $0.965 \pm 0.020$ | $0.831 \pm 0.032$ | $0.661 \pm 0.082$ |
| SCRUB | $0.988 \pm 0.010$ | $0.923 \pm 0.033$ | $0.780 \pm 0.101$ |
| Influence unlearning | $0.986 \pm 0.031$ | $0.738 \pm 0.051$ | $0.381 \pm 0.037$ |
| Salun | $0.774 \pm 0.043$ | $0.758 \pm 0.053$ | $0.886 \pm 0.082$ |
| Random-label | $0.709 \pm 0.029$ | $0.548 \pm 0.093$ | $0.877 \pm 0.064$ |

(a) CIFAR-10 with ResNet-18

|  | Low memorization | Medium memorization | High memorization |
|---|---|---|---|
| Retrain | $1.000 \pm 0.000$ | $1.000 \pm 0.000$ | $1.000 \pm 0.000$ |
| Original | $0.983 \pm 0.014$ | $0.516 \pm 0.041$ | $0.026 \pm 0.005$ |
| Fine-tune | $0.974 \pm 0.064$ | $0.830 \pm 0.052$ | $0.768 \pm 0.025$ |
| L1-sparse | $0.857 \pm 0.035$ | $0.828 \pm 0.021$ | $0.754 \pm 0.036$ |
| NegGrad | $0.680 \pm 0.130$ | $0.564 \pm 0.044$ | $0.270 \pm 0.039$ |
| NegGrad+ | $0.986 \pm 0.024$ | $0.917 \pm 0.040$ | $0.889 \pm 0.012$ |
| SCRUB | $0.982 \pm 0.017$ | $0.761 \pm 0.051$ | $0.594 \pm 0.118$ |
| Influence unlearning | $0.960 \pm 0.112$ | $0.556 \pm 0.044$ | $0.154 \pm 0.081$ |
| SalUn | $0.964 \pm 0.091$ | $0.564 \pm 0.047$ | $0.538 \pm 0.061$ |
| Random-label | $0.770 \pm 0.105$ | $0.548 \pm 0.022$ | $0.409 \pm 0.044$ |

(b) CIFAR-100 with ResNet-50

Table 12: Accuracy and MIA results for various unlearning algorithms applied on forget / retain sets with different memorization levels for CIFAR10 using ResNet-18.

|  | Forget accuracy | Retain accuracy | Test accuracy | MIA |
|---|---|---|---|---|
| Retrain | 99.711 ± 0.253 | 100.000 ± 0.000 | 84.280 ± 1.184 | 0.010 ± 0.006 |
| Original | 100.000 ± 0.000 | 100.000 ± 0.000 | 84.353 ± 3.455 | 0.000 ± 0.000 |
| Fine-tune | 99.156 ± 1.634 | 98.390 ± 1.731 | 79.587 ± 3.173 | 0.020 ± 0.024 |
| L1-sparse | 98.678 ± 2.046 | 97.736 ± 0.449 | 78.750 ± 4.659 | 0.039 ± 0.047 |
| NegGrad | 99.178 ± 1.574 | 99.241 ± 1.512 | 79.390 ± 1.904 | 0.030 ± 0.033 |
| NegGrad+ | 98.300 ± 1.490 | 99.944 ± 0.142 | 82.257 ± 3.363 | 0.020 ± 0.019 |
| SCRUB | 99.911 ± 0.126 | 100.000 ± 0.000 | 84.323 ± 3.794 | 0.019 ± 0.006 |
| Influence unlearning | 100.000 ± 0.000 | 100.000 ± 0.000 | 83.133 ± 4.128 | 0.002 ± 0.004 |
| SalUn | 81.278 ± 5.906 | 99.996 ± 0.003 | 79.223 ± 5.991 | 0.960 ± 0.053 |
| Random-label | 86.667 ± 6.902 | 90.604 ± 7.740 | 74.350 ± 1.287 | 0.611 ± 0.089 |

(a) Low memorization

|  | Forget accuracy | Retain accuracy | Test accuracy | MIA |
|---|---|---|---|---|
| Retrain | 73.611 ± 3.478 | 100.000 ± 0.000 | 82.977 ± 2.060 | 0.654 ± 0.067 |
| Original | 100.000 ± 0.000 | 100.000 ± 0.000 | 84.353 ± 3.455 | 0.000 ± 0.000 |
| Fine-tune | 82.856 ± 1.706 | 99.422 ± 0.721 | 80.957 ± 2.376 | 0.413 ± 0.117 |
| L1-sparse | 82.922 ± 5.971 | 95.660 ± 5.166 | 77.130 ± 4.536 | 0.368 ± 0.013 |
| NegGrad | 92.900 ± 10.384 | 96.755 ± 6.091 | 77.593 ± 5.946 | 0.164 ± 0.130 |
| NegGrad+ | 88.122 ± 5.826 | 99.962 ± 0.072 | 80.303 ± 5.202 | 0.193 ± 0.060 |
| SCRUB | 69.411 ± 18.532 | 99.963 ± 0.127 | 81.787 ± 4.574 | 0.582 ± 0.140 |
| Influence unlearning | 91.800 ± 13.743 | 96.267 ± 6.256 | 76.843 ± 5.060 | 0.217 ± 0.099 |
| SalUn | 51.100 ± 7.729 | 100.000 ± 0.000 | 80.847 ± 3.332 | 0.965 ± 0.012 |
| Random-label | 36.233 ± 6.904 | 94.675 ± 2.471 | 75.373 ± 2.546 | 0.829 ± 0.076 |

(b) Medium memorization

|  | Forget accuracy | Retain accuracy | Test accuracy | MIA |
|---|---|---|---|---|
| Retrain | 23.444 ± 5.753 | 100.000 ± 0.000 | 82.967 ± 1.910 | 0.961 ± 0.018 |
| Original | 100.000 ± 0.000 | 100.000 ± 0.000 | 84.353 ± 3.455 | 0.002 ± 0.001 |
| Fine-tune | 45.822 ± 10.494 | 98.987 ± 4.112 | 82.003 ± 4.700 | 0.820 ± 0.135 |
| L1-sparse | 55.644 ± 16.226 | 97.001 ± 1.941 | 78.730 ± 5.902 | 0.714 ± 0.075 |
| NegGrad | 85.133 ± 15.675 | 93.289 ± 10.807 | 74.827 ± 7.390 | 0.266 ± 0.179 |
| NegGrad+ | 48.400 ± 5.596 | 94.876 ± 2.829 | 75.783 ± 4.310 | 0.641 ± 0.107 |
| SCRUB | 44.167 ± 11.976 | 99.916 ± 0.341 | 82.317 ± 5.227 | 0.840 ± 0.083 |
| Influence unlearning | 74.567 ± 3.934 | 88.536 ± 2.176 | 71.143 ± 3.499 | 0.417 ± 0.067 |
| SalUn | 32.311 ± 5.235 | 99.560 ± 0.597 | 80.647 ± 3.328 | 0.951 ± 0.008 |
| Random-label | 22.700 ± 3.360 | 94.728 ± 4.453 | 76.323 ± 2.830 | 0.936 ± 0.025 |

(c) High memorization

Table 13: Accuracy and MIA performance for different unlearning algorithms applied on forget / retain sets of varying levels of memorization for CIFAR100 using ResNet-50.

|  | Forget accuracy | Retain accuracy | Test accuracy | MIA |
|---|---|---|---|---|
| Retrain | $99.878 \pm 0.096$ | $99.960 \pm 0.003$ | $74.077 \pm 2.112$ | $0.016 \pm 0.005$ |
| Original | $100.000 \pm 0.000$ | $99.959 \pm 0.015$ | $75.003 \pm 2.809$ | $0.001 \pm 0.001$ |
| Fine-tune | $99.878 \pm 0.266$ | $99.806 \pm 0.273$ | $71.693 \pm 4.515$ | $0.018 \pm 0.015$ |
| L1-sparse | $98.100 \pm 0.299$ | $95.385 \pm 2.147$ | $65.550 \pm 2.179$ | $0.040 \pm 0.007$ |
| NegGrad | $90.589 \pm 2.987$ | $85.454 \pm 6.778$ | $61.637 \pm 6.310$ | $0.140 \pm 0.010$ |
| NegGrad+ | $99.556 \pm 0.345$ | $99.951 \pm 0.018$ | $73.823 \pm 2.673$ | $0.014 \pm 0.010$ |
| SCRUB | $99.933 \pm 0.143$ | $99.971 \pm 0.014$ | $75.127 \pm 3.093$ | $0.003 \pm 0.006$ |
| Influence unlearning | $99.867 \pm 0.430$ | $99.077 \pm 2.993$ | $71.843 \pm 9.367$ | $0.006 \pm 0.017$ |
| Salun | $99.922 \pm 0.191$ | $99.225 \pm 1.600$ | $71.227 \pm 6.693$ | $0.008 \pm 0.008$ |
| Random-label | $98.144 \pm 1.559$ | $88.833 \pm 5.531$ | $62.207 \pm 4.559$ | $0.030 \pm 0.011$ |

(a) Low memorization

|  | Forget accuracy | Retain accuracy | Test accuracy | MIA |
|---|---|---|---|---|
| Retrain | $52.622 \pm 4.204$ | $99.976 \pm 0.006$ | $72.767 \pm 2.651$ | $0.906 \pm 0.015$ |
| Original | $99.800 \pm 0.000$ | $99.973 \pm 0.015$ | $75.003 \pm 2.809$ | $0.020 \pm 0.007$ |
| Fine-tune | $60.178 \pm 14.403$ | $96.944 \pm 5.329$ | $65.433 \pm 7.219$ | $0.663 \pm 0.102$ |
| L1-sparse | $56.567 \pm 3.067$ | $94.525 \pm 3.186$ | $64.013 \pm 1.340$ | $0.615 \pm 0.028$ |
| NegGrad | $82.000 \pm 4.362$ | $87.858 \pm 4.205$ | $63.673 \pm 5.832$ | $0.294 \pm 0.062$ |
| NegGrad+ | $54.267 \pm 14.532$ | $99.566 \pm 0.704$ | $69.440 \pm 4.251$ | $0.649 \pm 0.161$ |
| SCRUB | $74.511 \pm 9.443$ | $99.404 \pm 2.483$ | $72.787 \pm 8.967$ | $0.576 \pm 0.032$ |
| Influence unlearning | $84.667 \pm 15.692$ | $90.643 \pm 12.409$ | $63.330 \pm 11.045$ | $0.220 \pm 0.095$ |
| Salun | $79.889 \pm 8.212$ | $85.875 \pm 4.268$ | $63.147 \pm 1.900$ | $0.646 \pm 0.740$ |
| Random-label | $68.022 \pm 15.101$ | $77.440 \pm 12.416$ | $56.787 \pm 6.257$ | $0.354 \pm 0.133$ |

(b) Medium memorization

|  | Forget accuracy | Retain accuracy | Test accuracy | MIA |
|---|---|---|---|---|
| Retrain | $2.556 \pm 0.669$ | $99.972 \pm 0.009$ | $73.580 \pm 0.661$ | $1.000 \pm 0.001$ |
| Original | $99.900 \pm 0.166$ | $99.966 \pm 0.003$ | $75.003 \pm 2.809$ | $0.045 \pm 0.020$ |
| Fine-tune | $12.300 \pm 2.010$ | $94.729 \pm 1.488$ | $63.413 \pm 3.273$ | $0.947 \pm 0.003$ |
| L1-sparse | $15.244 \pm 2.825$ | $94.814 \pm 4.718$ | $64.707 \pm 2.808$ | $0.940 \pm 0.023$ |
| NegGrad | $62.578 \pm 14.734$ | $80.621 \pm 9.071$ | $58.027 \pm 9.041$ | $0.493 \pm 0.096$ |
| NegGrad+ | $5.567 \pm 2.654$ | $97.193 \pm 2.041$ | $67.820 \pm 0.927$ | $0.978 \pm 0.005$ |
| SCRUB | $40.900 \pm 20.728$ | $99.086 \pm 3.232$ | $71.220 \pm 9.090$ | $0.899 \pm 0.030$ |
| Influence unlearning | $85.389 \pm 9.154$ | $95.745 \pm 2.685$ | $67.240 \pm 1.869$ | $0.331 \pm 0.170$ |
| Salun | $33.556 \pm 6.167$ | $88.021 \pm 2.044$ | $62.127 \pm 2.447$ | $0.997 \pm 0.004$ |
| Random-label | $39.778 \pm 13.921$ | $78.972 \pm 8.172$ | $56.333 \pm 3.225$ | $0.645 \pm 0.101$ |

(c) High memorization

Table 14: Unlearning performance evaluated by ToW-MIA, with ToW results included for comparison. **Subtables (a, b)**: ToW-MIA for different unlearning algorithms applied to forget / retain sets with varying ES or memorization levels on CIFAR-10 using ResNet-18, with ToW results from Tables 6 and 11 for comparison. **Subtable (c)**: RUM$^{\mathcal{F}}$ and RUM results evaluated by ToW-MIA on CIFAR-10 using ResNet-18, with ToW results from Table 16 for comparison.

| | ToW-MIA | | | ToW | | |
| | Low ES | Medium ES | High ES | Low ES | Medium ES | High ES |
|---|---|---|---|---|---|---|
| Retrain | $1.000 \pm 0.000$ | $1.000 \pm 0.000$ | $1.000 \pm 0.000$ | $1.000 \pm 0.000$ | $1.000 \pm 0.000$ | $1.000 \pm 0.000$ |
| Original | $0.893 \pm 0.014$ | $0.841 \pm 0.026$ | $0.430 \pm 0.035$ | $0.944 \pm 0.014$ | $0.928 \pm 0.022$ | $0.759 \pm 0.048$ |
| Fine-tune | $0.921 \pm 0.061$ | $0.882 \pm 0.020$ | $0.768 \pm 0.055$ | $0.952 \pm 0.024$ | $0.923 \pm 0.019$ | $0.908 \pm 0.018$ |
| L1-sparse | $0.895 \pm 0.014$ | $0.845 \pm 0.031$ | $0.706 \pm 0.103$ | $0.945 \pm 0.013$ | $0.926 \pm 0.019$ | $0.836 \pm 0.031$ |
| NegGrad | $0.883 \pm 0.030$ | $0.841 \pm 0.044$ | $0.479 \pm 0.010$ | $0.929 \pm 0.030$ | $0.887 \pm 0.047$ | $0.766 \pm 0.042$ |
| NegGrad+ | $0.893 \pm 0.026$ | $0.877 \pm 0.057$ | $0.550 \pm 0.055$ | $0.941 \pm 0.029$ | $0.920 \pm 0.038$ | $0.800 \pm 0.029$ |
| SCRUB | $0.895 \pm 0.014$ | $0.884 \pm 0.045$ | $0.778 \pm 0.075$ | $0.944 \pm 0.014$ | $0.939 \pm 0.021$ | $0.920 \pm 0.033$ |
| Influence unlearning | $0.879 \pm 0.022$ | $0.812 \pm 0.020$ | $0.618 \pm 0.010$ | $0.928 \pm 0.023$ | $0.890 \pm 0.012$ | $0.744 \pm 0.029$ |
| SalUn | $0.339 \pm 0.137$ | $0.359 \pm 0.178$ | $0.597 \pm 0.126$ | $0.783 \pm 0.031$ | $0.656 \pm 0.046$ | $0.618 \pm 0.056$ |
| Random-label | $0.171 \pm 0.037$ | $0.446 \pm 0.219$ | $0.557 \pm 0.086$ | $0.767 \pm 0.107$ | $0.663 \pm 0.055$ | $0.443 \pm 0.087$ |

(a) ToW-MIA vs ES

| | ToW-MIA | | | ToW | | |
| | Low mem | Medium mem | High mem | Low mem | Medium mem | High mem |
|---|---|---|---|---|---|---|
| Retrain | $1.000 \pm 0.000$ | $1.000 \pm 0.000$ | $1.000 \pm 0.000$ | $1.000 \pm 0.000$ | $1.000 \pm 0.000$ | $1.000 \pm 0.000$ |
| Original | $0.973 \pm 0.037$ | $0.340 \pm 0.072$ | $0.041 \pm 0.017$ | $0.988 \pm 0.007$ | $0.723 \pm 0.053$ | $0.231 \pm 0.058$ |
| Fine-tune | $0.935 \pm 0.032$ | $0.842 \pm 0.037$ | $0.738 \pm 0.068$ | $0.933 \pm 0.052$ | $0.884 \pm 0.019$ | $0.760 \pm 0.065$ |
| L1-sparse | $0.904 \pm 0.098$ | $0.700 \pm 0.018$ | $0.642 \pm 0.025$ | $0.914 \pm 0.061$ | $0.816 \pm 0.011$ | $0.629 \pm 0.087$ |
| NegGrad | $0.933 \pm 0.054$ | $0.466 \pm 0.024$ | $0.258 \pm 0.105$ | $0.938 \pm 0.028$ | $0.738 \pm 0.005$ | $0.325 \pm 0.098$ |
| NegGrad+ | $0.973 \pm 0.028$ | $0.523 \pm 0.037$ | $0.600 \pm 0.123$ | $0.965 \pm 0.020$ | $0.831 \pm 0.032$ | $0.661 \pm 0.082$ |
| SCRUB | $0.979 \pm 0.011$ | $0.913 \pm 0.158$ | $0.865 \pm 0.050$ | $0.988 \pm 0.010$ | $0.923 \pm 0.033$ | $0.780 \pm 0.101$ |
| Influence unlearning | $0.973 \pm 0.030$ | $0.507 \pm 0.046$ | $0.357 \pm 0.061$ | $0.986 \pm 0.031$ | $0.738 \pm 0.051$ | $0.381 \pm 0.037$ |
| SalUn | $0.054 \pm 0.030$ | $0.674 \pm 0.097$ | $0.963 \pm 0.024$ | $0.774 \pm 0.043$ | $0.758 \pm 0.053$ | $0.886 \pm 0.082$ |
| Random-label | $0.328 \pm 0.394$ | $0.722 \pm 0.119$ | $0.862 \pm 0.084$ | $0.709 \pm 0.029$ | $0.548 \pm 0.093$ | $0.877 \pm 0.064$ |

(b) ToW-MIA vs memorization

| | ToW-MIA | ToW |
|---|---|---|
| Retrain | $1.000 \pm 0.000$ | $1.000 \pm 1.000$ |
| Fine-tune | $0.820 \pm 0.042$ | $0.849 \pm 0.030$ |
| Fine-tune shuffle | $0.848 \pm 0.125$ | $0.712 \pm 0.040$ |
| Fine-tune RUM$^{\mathcal{F}}$ | $0.888 \pm 0.022$ | $0.937 \pm 0.052$ |
| SalUn | $0.602 \pm 0.019$ | $0.731 \pm 0.070$ |
| SalUn shuffle | $0.677 \pm 0.023$ | $0.727 \pm 0.030$ |
| SalUn RUM$^{\mathcal{F}}$ | $0.878 \pm 0.049$ | $0.887 \pm 0.069$ |
| NegGrad+ | $0.707 \pm 0.033$ | $0.802 \pm 0.028$ |
| NegGrad+ shuffle | $0.470 \pm 0.027$ | $0.632 \pm 0.022$ |
| NegGrad+ RUM$^{\mathcal{F}}$ | $0.779 \pm 0.025$ | $0.879 \pm 0.068$ |
| L1-sparse | $0.772 \pm 0.014$ | $0.794 \pm 0.035$ |
| L1-sparse shuffle | $0.624 \pm 0.039$ | $0.716 \pm 0.023$ |
| L1-sparse RUM$^{\mathcal{F}}$ | $0.855 \pm 0.023$ | $0.900 \pm 0.020$ |
| Nothing→Fine-tune→SalUn | $0.950 \pm 0.062$ | $0.965 \pm 0.014$ |

(c) ToW-MIA vs RUM

Table 15: Class distribution of the forget set in the RUM experiment for CIFAR-10 and CIFAR-100, with 3000 examples in the forget set for each dataset. The forget set includes examples from all classes in both datasets.

| Class | 0 | 1 | 2 | 3 | 4 | 5 | 6 | 7 | 8 | 9 |
|---|---|---|---|---|---|---|---|---|---|---|
| Count | 291 | 259 | 363 | 314 | 277 | 316 | 298 | 309 | 306 | 267 |

(a) CIFAR-10

| Class | 0 | 1 | 2 | 3 | 4 | 5 | 6 | 7 | 8 | 9 | 10 | 11 | 12 | 13 | 14 | 15 | 16 | 17 | 18 | 19 |
|---|---|---|---|---|---|---|---|---|---|---|---|---|---|---|---|---|---|---|---|---|
| Count | 23 | 24 | 34 | 29 | 34 | 25 | 25 | 40 | 17 | 43 | 34 | 30 | 27 | 21 | 25 | 30 | 25 | 32 | 37 | 26 |
| Class | 20 | 21 | 22 | 23 | 24 | 25 | 26 | 27 | 28 | 29 | 30 | 31 | 32 | 33 | 34 | 35 | 36 | 37 | 38 | 39 |
| Count | 31 | 22 | 29 | 30 | 39 | 37 | 48 | 30 | 33 | 30 | 25 | 27 | 45 | 40 | 31 | 23 | 28 | 30 | 32 | 31 |
| Class | 40 | 41 | 42 | 43 | 44 | 45 | 46 | 47 | 48 | 49 | 50 | 51 | 52 | 53 | 54 | 55 | 56 | 57 | 58 | 59 |
| Count | 32 | 32 | 36 | 27 | 28 | 26 | 28 | 32 | 34 | 22 | 38 | 32 | 19 | 34 | 20 | 41 | 29 | 40 | 20 | 34 |
| Class | 60 | 61 | 62 | 63 | 64 | 65 | 66 | 67 | 68 | 69 | 70 | 71 | 72 | 73 | 74 | 75 | 76 | 77 | 78 | 79 |
| Count | 28 | 26 | 21 | 33 | 31 | 32 | 25 | 35 | 26 | 28 | 19 | 19 | 37 | 26 | 53 | 33 | 20 | 23 | 34 | 37 |
| Class | 80 | 81 | 82 | 83 | 84 | 85 | 86 | 87 | 88 | 89 | 90 | 91 | 92 | 93 | 94 | 95 | 96 | 97 | 98 | 99 |
| Count | 24 | 22 | 27 | 24 | 28 | 21 | 27 | 34 | 32 | 27 | 22 | 38 | 32 | 39 | 43 | 30 | 41 | 28 | 24 | 25 |

(b) CIFAR-100

Table 16: RUM results on CIFAR-10 and CIFAR-100. Results obtained by applying RUM with different algorithms, according to ToW (higher is better), its constituent ingredients, and MIA (lower is better for MIA gap). The **top section** compares applying an unlearning algorithm $\mathcal{U}$ in three ways: i) in one-go, as usual (e.g. Fine-tune), ii) on three randomly-determined equal-sized subsets of $\mathcal{S}$, sequentially (e.g. Fine-tune shuffle), and iii) on three equal-sized buckets obtained by refinement $\mathcal{F}(\mathcal{S})$ according to memorization scores, in low $\rightarrow$ med $\rightarrow$ high order (e.g. Fine-tune RUM$^{\mathcal{F}}$). The **middle section** of Table 16a further experiments with picking a different unlearning algorithm for each subset of $\mathcal{F}(\mathcal{S})$. Here A $\rightarrow$ B $\rightarrow$ C denotes applying algorithm A on the first subset, B on the second subset, and C on the third subset, where the subsets appear in low $\rightarrow$ medium $\rightarrow$ high order. The **bottom section** shows different orderings.

| | ToW (↑) | Forget accuracy | Retain accracy | Test accracy | MIA | MIA gap (↓) |
|---|---|---|---|---|---|---|
| Retrain | 1.000±0.000 | 64.156±2.632 | 100.000±0.000 | 83.917±2.040 | 0.549±0.020 | 0.000 |
| Fine-tune | 0.849±0.030 | 73.067±14.064 | 97.923±4.649 | 79.277±7.487 | 0.429±0.086 | 0.120 |
| Fine-tune shuffle | 0.712±0.040 | 88.389±9.089 | 96.578±8.717 | 82.037±8.500 | 0.451±0.037 | 0.098 |
| Fine-tune RUM$^{\mathcal{F}}$ | 0.937±0.052 | 69.133±6.583 | 99.664±0.175 | 83.230±2.221 | 0.450±0.038 | 0.099 |
| L1-sparse | 0.794±0.035 | 79.300±11.467 | 98.187±2.695 | 79.373±6.872 | 0.374±0.097 | 0.175 |
| L1-sparse shuffle | 0.716±0.023 | 88.244±5.217 | 97.152±1.462 | 81.047±3.378 | 0.211±0.077 | 0.338 |
| L1-sparse RUM$^{\mathcal{F}}$ | 0.900±0.020 | 69.967±6.014 | 98.458±1.399 | 81.047±3.378 | 0.477±0.039 | 0.072 |
| NegGrad+ | 0.802±0.028 | 76.867±7.382 | 98.415±0.783 | 77.337±4.664 | 0.319±0.093 | 0.230 |
| NegGrad+ shuffle | 0.632±0.022 | 99.600±0.933 | 99.929±0.141 | 83.700±5.336 | 0.029±0.029 | 0.520 |
| NegGrad+ RUM$^{\mathcal{F}}$ | 0.879±0.068 | 61.744±2.097 | 96.655±2.274 | 77.063±1.991 | 0.415±0.027 | 0.134 |
| SalUn | 0.731±0.070 | 40.056±0.391 | 99.796±0.429 | 80.370±4.159 | 0.923±0.067 | 0.374 |
| SalUn shuffle | 0.727±0.030 | 81.889±1.528 | 94.693±2.866 | 77.270±0.390 | 0.315±0.019 | 0.234 |
| SalUn RUM$^{\mathcal{F}}$ | 0.887±0.069 | 62.878±3.726 | 96.457±2.482 | 77.857±3.280 | 0.518±0.060 | 0.031 |
| nothing $\rightarrow$ Fine-tune $\rightarrow$ SalUn | 0.965±0.014 | 66.011±5.139 | 99.205±1.962 | 83.007±3.941 | 0.515±0.019 | 0.034 |
| nothing $\rightarrow$ SalUn $\rightarrow$ Fine-tune | 0.911±0.010 | 69.322±6.915 | 99.070±2.281 | 81.457±6.968 | 0.440±0.047 | 0.109 |
| nothing $\rightarrow$ SalUn $\rightarrow$ SalUn | 0.919±0.059 | 70.733±4.310 | 100.000±0.000 | 84.190±3.328 | 0.644±0.010 | 0.095 |
| Fine-tune RUM (low $\rightarrow$ med $\rightarrow$ high) | 0.937±0.052 | 69.133±6.583 | 99.664±0.175 | 83.230±2.221 | 0.450±0.038 | 0.099 |
| Fine-tune RUM (high $\rightarrow$ med $\rightarrow$ low) | 0.942±0.032 | 68.222±1.184 | 99.515±2.020 | 83.587±2.236 | 0.478±0.120 | 0.071 |
| SalUn RUM (low $\rightarrow$ med $\rightarrow$ high) | 0.887±0.069 | 62.878±3.726 | 96.457±2.482 | 77.857±3.280 | 0.518±0.060 | 0.031 |
| SalUn RUM (high $\rightarrow$ med $\rightarrow$ low) | 0.881±0.024 | 74.644±1.615 | 99.921±0.124 | 82.393±3.018 | 0.328±0.011 | 0.221 |

(a) RUM results on CIFAR-10 using ResNet-18

| | ToW (↑) | Forget accuracy | Retain accracy | Test accracy | MIA | avg. MIA gap (↓) |
|---|---|---|---|---|---|---|
| Retrain | 1.000±0.000 | 52.044±2.160 | 99.966±0.018 | 73.260±2.150 | 0.635±0.005 | 0.000 |
| NegGrad+ | 0.861±0.069 | 63.644±4.508 | 99.641±0.316 | 70.970±3.014 | 0.477±0.019 | 0.159 |
| NegGrad+ shuffle | 0.613±0.054 | 88.011±4.628 | 97.994±1.394 | 70.867±2.836 | 0.218±0.074 | 0.417 |
| NegGrad+ RUM$^{\mathcal{F}}$ | 0.921±0.034 | 50.789±7.713 | 98.413±0.855 | 68.820±3.633 | 0.576±0.075 | 0.059 |
| L1-sparse | 0.824±0.011 | 54.078±3.275 | 93.367±1.376 | 63.287±2.381 | 0.546±0.028 | 0.089 |
| L1-sparse shuffle | 0.604±0.023 | 82.111±2.285 | 95.310±0.879 | 63.880±2.323 | 0.282±0.014 | 0.353 |
| L1-sparse RUM$^{\mathcal{F}}$ | 0.883±0.046 | 52.444±1.628 | 97.269±1.393 | 64.317±4.990 | 0.602±0.007 | 0.033 |
| Fine-tune | 0.734±0.025 | 76.400±4.445 | 99.575±0.761 | 70.740±3.421 | 0.496±0.156 | 0.139 |
| Fine-tune shuffle | 0.589±0.036 | 81.689±5.860 | 93.616±2.316 | 62.653±3.413 | 0.290±0.049 | 0.345 |
| Fine-tune RUM$^{\mathcal{F}}$ | 0.784±0.040 | 61.767±10.301 | 96.706±3.397 | 63.090±3.575 | 0.542±0.030 | 0.093 |
| SalUn | 0.545±0.061 | 88.967±28.001 | 94.207±19.407 | 66.327±10.914 | 0.259±0.052 | 0.372 |
| SalUn shuffle | 0.538±0.019 | 63.389±3.817 | 75.479±3.208 | 53.713±2.518 | 0.398±0.022 | 0.237 |
| SalUn RUM$^{\mathcal{F}}$ | 0.614±0.037 | 58.489±1.581 | 79.719±1.225 | 55.607±3.857 | 0.454±0.008 | 0.181 |
| NegGrad+ RUM (low $\rightarrow$ med $\rightarrow$ high) | 0.921±0.034 | 50.789±7.713 | 98.413±0.855 | 68.820±3.633 | 0.576±0.075 | 0.059 |
| NegGrad+ RUM (high $\rightarrow$ med $\rightarrow$ low) | 0.929±0.058 | 49.900±12.333 | 99.703±0.340 | 71.080±3.141 | 0.657±0.040 | 0.022 |
| L1-sparse RUM (low $\rightarrow$ med $\rightarrow$ high) | 0.883±0.046 | 52.444±1.628 | 97.269±1.393 | 64.317±4.990 | 0.602±0.007 | 0.033 |
| L1-sparse RUM (high $\rightarrow$ med $\rightarrow$ low) | 0.908±0.013 | 53.967±2.587 | 98.732±1.376 | 66.963±2.814 | 0.591±0.020 | 0.044 |

(b) RUM results on CIFAR-100 using ResNet-50

Table 17: RUM results using C-proxy for the refinement step, evaluated by ToW on CIFAR-10 and Tiny-ImageNet datasets. Each algorithm is applied in three ways: RUM$^{\mathcal{F}}$, as well as vanilla and shuffle as comparison. Results are averaged over 3 runs with 95% confidence intervals.

| | ToW ($\uparrow$) | Forget accuracy | Retain accracy | Test accracy | MIA | MIA gap ($\downarrow$) |
|---|---|---|---|---|---|---|
| Retrain | 1.000±0.000 | 50.433±6.808 | 100.000±0.000 | 84.167±1.616 | 0.637±0.032 | 0.000 |
| Fine-tune | 0.731±0.021 | 77.389±3.298 | 97.443±2.263 | 81.847±1.021 | 0.286±0.062 | 0.351 |
| Fine-tune shuffle | 0.829±0.022 | 62.800±16.311 | 98.060±5.309 | 80.677±5.082 | 0.560±0.091 | 0.077 |
| Fine-tune RUM$^{\mathcal{F}}$ | 0.927±0.047 | 58.400±1.723 | 98.920±2.484 | 82.007±3.186 | 0.482±0.004 | 0.155 |
| L1-sparse | 0.754±0.071 | 66.856±9.041 | 96.095±4.676 | 78.200±5.173 | 0.492±0.041 | 0.145 |
| L1-sparse shuffle | 0.618±0.095 | 83.222±1.391 | 95.890±4.480 | 79.963±5.127 | 0.256±0.031 | 0.381 |
| L1-sparse RUM$^{\mathcal{F}}$ | 0.907±0.026 | 53.611±3.446 | 96.947±1.991 | 80.783±2.540 | 0.566±0.021 | 0.071 |
| NegGrad+ | 0.880±0.039 | 56.067±5.261 | 99.037±1.532 | 78.443±5.531 | 0.504±0.049 | 0.133 |
| NegGrad+ shuffle | 0.529±0.071 | 94.867±9.081 | 98.919±2.120 | 80.563±6.630 | 0.110±0.107 | 0.527 |
| NegGrad+ RUM$^{\mathcal{F}}$ | 0.893±0.026 | 58.400±1.723 | 98.920±2.484 | 82.007±3.186 | 0.482±0.004 | 0.155 |
| SalUn | 0.859±0.042 | 58.111±4.816 | 98.030±1.356 | 79.180±4.306 | 0.593±0.014 | 0.044 |
| SalUn shuffle | 0.638±0.071 | 84.067±5.039 | 97.897±0.449 | 82.387±3.331 | 0.381±0.015 | 0.256 |
| SalUn RUM$^{\mathcal{F}}$ | 0.737±0.040 | 75.578±4.606 | 99.996±0.009 | 82.597±4.474 | 0.949±0.049 | 0.312 |
| SCRUB vanilla | 0.802±0.047 | 71.733±2.630 | 99.671±0.708 | 82.077±4.319 | 0.462±0.031 | 0.175 |
| SCRUB shuffle | 0.732±0.059 | 74.678±20.089 | 95.617±11.622 | 80.983±8.000 | 0.325±0.163 | 0.312 |
| SCRUB RUM$^{\mathcal{F}}$ | 0.945±0.017 | 50.589±0.526 | 99.375±0.560 | 82.487±3.277 | 0.500±0.003 | 0.137 |

(a) ToW vs RUM on CIFAR-10 with ResNet-18

| | ToW ($\uparrow$) | Forget accuracy | Retain accracy | Test accracy | MIA | MIA gap ($\downarrow$) |
|---|---|---|---|---|---|---|
| Retrain | 1.000±0.000 | 45.633±2.521 | 99.999±0.001 | 64.606±0.230 | 0.662±0.005 | 0.000 |
| Fine-tune | 0.777±0.020 | 52.356±3.990 | 91.768±2.396 | 55.438±1.704 | 0.530±0.024 | 0.132 |
| Fine-tune shuffle | 0.566±0.021 | 80.622±0.981 | 94.195±0.645 | 56.971±1.295 | 0.276±0.026 | 0.386 |
| Fine-tune RUM$^{\mathcal{F}}$ | 0.803±0.023 | 51.578±3.275 | 93.174±1.606 | 56.185±1.393 | 0.554±0.032 | 0.108 |
| L1-sparse | 0.772±0.017 | 51.622±2.343 | 89.556±1.293 | 56.258±0.965 | 0.534±0.025 | 0.128 |
| L1-sparse shuffle | 0.584±0.025 | 82.478±0.172 | 98.426±0.629 | 58.498±1.567 | 0.244±0.035 | 0.418 |
| L1-sparse RUM$^{\mathcal{F}}$ | 0.883±0.032 | 55.722±0.485 | 99.985±0.058 | 62.833±0.258 | 0.660±0.007 | 0.002 |
| NegGrad+ | 0.859±0.066 | 51.122±11.593 | 97.359±1.060 | 57.992±1.441 | 0.677±0.160 | 0.015 |
| NegGrad+ shuffle | 0.482±0.004 | 85.433±21.932 | 93.073±17.340 | 51.710±18.821 | 0.568±0.648 | 0.094 |
| NegGrad+ RUM$^{\mathcal{F}}$ | 0.935±0.040 | 46.944±4.332 | 99.666±0.056 | 59.952±1.194 | 0.621±0.019 | 0.041 |
| SalUn | 0.689±0.008 | 67.178±2.079 | 89.889±0.888 | 62.319±0.748 | 0.556±0.209 | 0.106 |
| SalUn shuffle | 0.553±0.032 | 65.333±5.584 | 80.933±7.695 | 49.750±3.141 | 0.397±0.030 | 0.265 |
| SalUn RUM$^{\mathcal{F}}$ | 0.736±0.027 | 54.867±3.589 | 86.485±2.309 | 63.151±2.853 | 0.560±0.024 | 0.102 |

(b) ToW vs RUM on Tiny-ImageNet with ResNet-18

| | ToW ($\uparrow$) | Forget accuracy | Retain accracy | Test accracy | MIA | MIA gap ($\downarrow$) |
|---|---|---|---|---|---|---|
| Retrain | 1.000±0.000 | 49.100±1.910 | 99.995±0.003 | 60.699±0.207 | 0.637±0.013 | 0.000 |
| NegGrad+ | 0.637±0.106 | 79.356±31.411 | 97.146±11.969 | 55.678±12.037 | 0.301±0.188 | 0.336 |
| NegGrad+ shuffle | 0.514±0.009 | 78.078±42.146 | 86.404±29.465 | 47.103±25.866 | 0.376±0.598 | 0.261 |
| NegGrad+ RUM$^{\mathcal{F}}$ | 0.812±0.053 | 57.233±9.692 | 97.515±9.221 | 51.430±6.930 | 0.477±0.062 | 0.160 |
| L1-sparse | 0.771±0.030 | 53.167±2.384 | 89.907±1.114 | 50.117±1.047 | 0.500±0.031 | 0.137 |
| L1-sparse shuffle | 0.589±0.028 | 83.778±0.539 | 96.793±0.110 | 53.924±1.024 | 0.217±0.025 | 0.420 |
| L1-sparse RUM$^{\mathcal{F}}$ | 0.885±0.024 | 56.678±1.315 | 99.932±0.080 | 56.471±2.238 | 0.585±0.024 | 0.052 |
| Fine-tune | 0.750±0.013 | 49.433±1.168 | 87.214±2.139 | 47.343±1.151 | 0.528±0.016 | 0.109 |
| Fine-tune shuffle | 0.576±0.061 | 65.978±11.740 | 81.317±8.350 | 46.023±4.367 | 0.360±0.160 | 0.277 |
| Fine-tune RUM$^{\mathcal{F}}$ | 0.767±0.044 | 49.989±2.632 | 90.010±3.150 | 46.896±2.862 | 0.527±0.024 | 0.110 |
| SalUn | 0.602±0.041 | 85.522±2.569 | 98.336±0.254 | 56.945±1.357 | 0.322±0.035 | 0.315 |
| SalUn shuffle | 0.566±0.011 | 72.622±43.082 | 84.977±36.200 | 50.430±15.871 | 0.313±0.380 | 0.324 |
| SalUn RUM$^{\mathcal{F}}$ | 0.679±0.025 | 54.622±5.034 | 82.377±2.398 | 47.990±1.540 | 0.474±0.136 | 0.163 |

(c) ToW vs RUM on Tiny-ImageNet with VGG-16

