# OpenReview forum: "What makes unlearning hard and what to do about it"
_NeurIPS.cc/2024/Conference — NeurIPS 2024 poster_

### Official Review · Reviewer_7Yis · 2024-06-26

**Soundness:** 2
**Presentation:** 3
**Contribution:** 3
**Rating:** 6
**Confidence:** 3

**Summary:**

This paper studies the difficulty of unlearning with respect to different unlearning sets. Specifically, the authors examine two factors, namely entanglement between forget and retain sets, and memorization. The authors further propose a framework called RUM which first divides the forget set into different subsets according to the memorization score and performs different unlearning algorithms to unlearn each subset.

**Strengths:**

I agree that studying the difficulty of unlearning different data (subset) is a very important problem and would help us evaluate current approximate unlearning algorithms. I generally enjoyed reading the paper, especificallly Section 4, where the 2 factors are nicely chosen and the experiments (figure 1) well demonstrate their effects. Specifically, the memorization part aligns with the study of data attribution and matches my intuition for approaching this problem.

**Weaknesses:**

I will merge the weaknesses and my questions in this part such that the authors can easily address each point during the rebuttal:

**[Memorization]**: according to equation (1) and Section 4.2, it appears to me the authors perform "leave-one-out" experiments, where they train a model without a target data sample and calculate the memorization score in (1). This is repeated for every training data point (50000 times for CIFAR-10) and it seems quite expensive (if training until convergence). My questions here are:
- Did you use any approximation in your experiments? How many epochs did you train? Can you show the computational cost?
- In Section 6, the refinement step is also performed based on memorization scores. Thus I believe the calculation of memorization score should be also considered for "efficiency" in Section 7? Given a large model (e.g., LLMs), and a large forgetting set, this process would be extremely expensive and may decrease the practicality of RUM.

**[MIA and ToW]**: The authors use ToW as the only evaluation in Section 4 and apply both MIA and ToW for Section 5. It is unclear to me why they omit MIA for evaluating unlearning difficulty. Is it because MIA may be a biased evaluation? Sequentially, for step 3 in RUM, will the selection of optimal unlearning algorithm be affected by considering MIA as well?

**[Transferability]**: RUM relies on the observation in Section 4 to select optimal unlearning algorithms. This comes handy as Section 4 and 6 both consider the same datasets. However, in practical scenarios, this is not possible and one needs to rely on the assumption that one unlearning algorithm is optimal across different datasets and model architectures. Is this assumption true?

**[Title and Section 4]**: The title of this paper and name of Section 4 "what makes unlearning hard" is misleading to me. This presentation suggests that the authors are studying: (1) unlearning is a hard problem; (2) why it is hard. However this is not true and they actually studied "the difficulty of unlearning regarding choice of forgetting sets". Specifically, the memorization subsection actually suggests different behaviors for different unlearning methods. I would suggest a more accurate delivery such as "on the difficulty of unlearning...".

**Overall**, I like the problem of interest in this paper and I appreciate the takeaways from Section 4. I think the observations help the community understand unlearning better. However, I don't believe RUM is a super practical framework based on the tremendous effort needed to perform refinement and the unknown behavior of transferability. If any of my speculations are wrong I would love to hear back from the authors.

**Questions:**

See above.

**Limitations:**

Limitations are addressed.

---

> ### Author Rebuttal · Authors · 2024-08-06
>
> **Re: “leave-one-out experiments” and “did you use any approximation?”**. Let us clarify the setup.
> While the reviewer is correct that Feldman's definition of memorisation implies a 'leave-one-out' training, in practice this evaluation is prohibitively expensive, as pointed out. As a more efficient empirical approximation, (Feldman and Zhang, 2020 “What neural networks memorize and why: discovering the long tail via influence estimation”) suggests performing 'leave-k-out' training and repeating it many times (e.g. 4k times). This way, for each data point, one can find a sufficiently large number of models that include that point in the training and a sufficiently large number of models that don’t (in the pool of 4k models). In Feldman's experiments, k is 30% of the training dataset size. We have followed the same recipe to calculate the memorisation scores and our results agree with Feldman's findings. Thank you for pointing this out. We will add these clarifications to a section in the appendix for clarity (and refer to it from the main paper).
>
> Further, in our new experiments in the rebuttal, we have considered using a proxy based on confidences that we find correlates well with memorization scores, leading to an efficient and practical recipe using RUM. Using this proxy is actually key as its computation requires orders of magnitude fewer models than Feldman’s memorization score, making our framework practical while still highly performant! Please see the “common response” for details. We hope that this line of experimentation and our thorough results also alleviate the concern of the reviewer that “the refinement step is also performed based on memorization scores” and “this process may be extremely expensive and may decrease the practicality of RUM”, since our findings using this cheap proxy align with those obtained using the true memorization scores.
>
> **Re: “how many epochs did you train? Can you show the computation cost”**. We trained for 30 epochs for cifar10. For cifar100, we used the public memorization scores (computed by Feldman et al) and used the same model architecture (resnet50) as Feldman’s for the experiments. Please also see section A.2 in the appendix for additional details.
>
> **Re: MIA and ToW**. The reviewer is correct that we could have repeated our analysis in Section 4 using MIAs, and accordingly we could have also based RUM’s selection of the unlearning algorithm per subset based on those findings. In our original experiments, we made the design decision of using ToW there due to its simplicity and compute-efficiency. We want to emphasize that, ultimately, we evaluate the algorithms we produce (RUM variants) using both ToW and MIA, and we find improvement on both fronts! We view this general improvement as evidence that ToW is a useful proxy for gauging how behaviours of different algorithms vary according to key characteristics.
>
> However, during the rebuttal, we have expanded our analysis to address this feedback. Specifically, we tried a variant coined “ToW-MIA” that replaces the accuracy gap on the forget set term with the MIA gap (see Section 6). The results shown in Figures 7-9 in the PDF clearly show the same trends with ToW-MIA as for ToW, both for analyses and RUM. We will add these to the paper.
>
> **Re: transferability and “one needs to rely on the assumption that one unlearning algorithm is optimal across different datasets and model architectures”**. It is true that our findings in this paper don’t inform us about the behaviour of any algorithm on any dataset (a limitation shared by any empirical paper). Understanding behaviours of algorithms across different unlearning problems is actually a major challenge that the community is faced with still, perhaps owing to unlearning being a young area of research. We view our work as a first step in uncovering key factors that practitioners can focus on when attempting to investigate the best algorithms to use in new contexts, which were not previously known.
>
> We also emphasize that, in the context of a single chosen algorithm, even applying only the refinement step of RUM (applying that given algorithm in a sequence on refined subsets of the forget set, according to characteristics we discover) can boost performance significantly. This is very important as it offers a direct way to leverage our findings to get a boost in performance even when in scenarios where we don’t know the optimal algorithm to choose per subset (and rely on using a single pre-chosen algorithm).
>
> Finally, we note that during the rebuttal we have investigated an additional dataset and architecture, showing that our main conclusions hold there too, as evidence for transferability. Please see the “common response” to all reviewers for more details.
>
> **Re: title and section 4**. When we say “what makes unlearning hard”, the word “what” refers to “what properties of the forget set, or the retain / forget partition”, but that is a little bit of a mouthful, so we simplified it. We believe our paper studies what makes the problem hard, through showing under which circumstances (in terms of ES and memorization scores) different algorithms struggle to perform well; and we view this as largely conveying the same message as “on the difficulty of unlearning”. Does this clarification help, or does the reviewer still feel that the title is a little misleading? We are happy to brainstorm more about this.
>
> **Overall**, we thank the reviewer for the thoughtful feedback. We actually believe that our new results (with C-proxy; see the “common response”, and with new a new dataset and architecture) as well as the discussion above addresses the reviewer’s central concerns about the practicality and transferability of RUM. We would love to hear the reviewer’s thoughts on this and discuss further if there are remaining concerns. Thanks!

---

> > ### Comment · Reviewer_7Yis · 2024-08-09
> >
> > Dear Authors,
> >
> >
> > Sorry for the late reply, I was pretty occupied for the last few days. I have carefully read your rebuttal and additional experiments. And here are my further comments:
> >
> > - [**Leave-one-out**]: Leave-k-out makes sense and the new C-proxy experiments look great to me. You mentioned there is an order of magnitude improvement over Feldman's method in terms of efficiency. Still, I am curious about the comparison between the running time of the two algorithms. Can you provide a rough estimate on the CIFAR-10/ResNet experiment?
> >
> > - [**What makes unlearning hard**]: I like this clarification but I still think the current title could be a bit over-simplified. My suggestion is something like "What properties make data hard to unlearn". This is not critical though and I can also accept the paper with its current title.
> >
> > In summary, all the other concerns are well addressed. While I still think the contribution of "what makes unlearning hard" is more significant than "what to do about it", I am willing to recommend acceptance. I will adjust my ratings after another round of discussion with the authors.
> >
> >
> > 7Yis

---

> > > ### Author Response · Authors · 2024-08-09
> > > **Addressing reviewer 7Yis new questions**
> > >
> > > Dear reviewer 7Yis, thanks a lot for your meaningful engagement. Please see below for answers to your questions:
> > >
> > > **Re: Leave-one-out... and “Can you provide a rough estimate on the CIFAR-10/ResNet experiment?”**
> > >
> > > Indeed, with C-proxy, running time is not an issue. Specifically, the running times of the two algorithms are as follows:
> > > - For computing Feldman’s memorization scores, we had to train 450 models! Training one model takes ca. 15 minutes. We used 4 GPUs, with the total time taking ca. **1,500 minutes**.
> > >
> > > - Computing C-proxy, on the other hand (which is done during the training of the original model) adds ca. 2.5 seconds per epoch, for a total of ca. **one minute of overhead**!
> > >
> > > This is a drastic difference in running times, showing negligible overhead for computing C-proxy, and negligible resource requirements. These facts allow us to claim that our contributions are practicable!
> > >
> > > **Re: What makes unlearning hard: I like this clarification … and … about the title**.
> > >
> > > We hope the above explanations help alleviate the original concerns about the practicability of our contributions (and the resulting concerns about the title).
> > >
> > > Again, thanks a lot for your positive feedback to our rebuttal. Please let us know if you have any other questions.

---

> > > > ### Author Response · Authors · 2024-08-13
> > > > **Final communication**
> > > >
> > > > Dear reviewer 7Yis,
> > > >
> > > > As the deadline is approaching, please let us know if there are any other concerns.
> > > >
> > > > Thank you.

---

### Official Review · Reviewer_HxTF · 2024-07-12

**Soundness:** 4
**Presentation:** 2
**Contribution:** 3
**Rating:** 7
**Confidence:** 4

**Summary:**

This paper looks at measurable qualities of forget/retain sets in machine unlearning contexts, which can make it difficult to 1) unlearn the forget set at all 2) unlearn the forget set without also unlearning the retain set. The paper puts forward well-reasoned metrics that get at entanglement/homogeneity in the embedding space for the forget and retain set, and also examine memorization as a possible factor for complicating machine unlearning. Based on the lessons from this, the paper suggests a meta-learning algorithm to figure out how to use existing unlearning algorithms to better unlearn the forget set.

**Strengths:**

This paper studies a very sensible set of questions with respect to qualities of the forget set that could make machine unlearning hard. The results (on the motivation side) are arguably expected, but someone needed to actually do the work to show this, and I'm really glad this paper did so (and did so very comprehensively, with respect to entanglement and memorization, and the ToW metric). Doing this / using these (perhaps expected) lessons to motivate the meta-learning RUM algorithm is also very principled, and also makes for a really tight paper. Without doing the first half fully, it would be hard to understand why the second half/ proposed algorithm is so well-motivated.

**Weaknesses:**

Overall, I think this paper makes excellent contributions to unlearning science. There is one thing I would have liked to see addressed in  more detail in the main paper, and I have some nit comments (that do not affect my score) that I think would improve readability. I organize my response here according to these distinctions, to (hopefully) make it clear which things could possibly lead to improvements in my score.

**Affects my score, engaging with this could lead to me increasing my score**

Given how expensive unlearning methods tend to be still, it would be great to see more discussion about efficiency. It is not clear to me entirely from the main paper if there are significant associated costs of using this meta-learning approach. These costs might be entirely worth the benefits presented (depending on the context of use, of course), but it would be really useful to include more on this in the main paper.

However, I also note that this paper packs in so much already in the page count, that I understand why some things had to be cut.
But this also led to other problems (which I understand are in tension with what I'm saying about including more details about efficiency): the quantity of presented results for a paper of this length has demanded really small (often difficult to read and process) tables and figures. The reason this is important is not just about readability. It takes a lot more time as a reader to unpack and verify the results in relation to the claims being made.

**Nit comments, do not have an impact on my score**

Math readability: Using different symbol classes for things of different types could help readability of the definitions a lot in this paper. For example, curly braces are used here for both algorithms and sets of datapoints. A simple improvement would be to use visual cues in the format to show they're different (not just with the chosen letter). For example, using bolded capitalization for a matrix of examples. Doing this can also make editing/finding typos more noticeable (e.g., line 60 has an $S$ for the forget set that, with the current style/format, should be $\mathcal{S}$).

Please explicitly define $e$ in the follow-on to Definition 2.1

**Questions:**

See weaknesses -- It would be great to understand more about efficiency and if there are interesting trade-offs here with respect to the proposed method

**Limitations:**

See above re: efficiency analysis

---

> ### Author Rebuttal · Authors · 2024-08-06
>
> **Re: More discussion about efficiency / costs of RUM**. This is a great point! Many researchers in this area do not bother with addressing issues of efficiency. Due to space constraints, we may have also been guilty of not paying as much attention as needed on this topic. This is unfortunate as indeed with our contributions we do address many issues about unlearning efficiency issues and those that arise as a result of our RUM framework.
>
> Please note the below, which we will add more thoroughly in the paper too:
>
> 1) Our contributions show the importance of memorization, how it affects different SOTA algorithms, and how it can be leveraged with our RUM framework. But computing memorization scores, a la Feldman, is a very computation-heavy process, requiring a large number (100s) of models. We only just alluded in the paper to using proxies for the Feldman memorization score. But we agree this requires more concrete discussion.
>
> In preliminary experiments, we experimented with various memorization score proxies from the literature. We have identified a promising one (C-proxy) and experimented with it in detail during the rebuttal (please see the “common response”). Our results show that: (i) our analyses and trends in Section 4 regarding the impact of memorization on unlearning algorithms continue to hold when using C-proxy instead, with no noticeable differences. (ii) leveraging the c-proxy values instead of the actual (a la Feldman) memorization scores still  leads to very large performance gains with our RUM framework, consistent with the ones in the original paper. In the attached PDF, please see Figure 1 and 2, showing the performance of algorithms according to low/medium/high c-proxy values and the performance of RUM when using C-proxy, respectively. As before the gains are large and clear with C-proxy as well!
>
> Please note that computing c-proxy instead of the Feldman’s memorization score enjoys orders of magnitude performance improvement  (due to avoiding training a large number of models that are needed to compute Feldman’s memorization scores). This makes our RUM framework practical and efficient to deploy.
>
> 2) Our contributions show when it makes sense to apply any of the SOTA algorithms, based on the memorization scores of examples in the forget set. So, a forget set consisting of only low-memorized examples, can be excluded from the unlearning process, saving all unlearning costs: as discussed in the paper, unlearning in this case is at best not needed (and at worse harmful), because even a model retrained without such examples can classify them correctly, so there is no need to modify the model to make it unable to classify those examples. In fact, attempts at doing so may lead to worse unlearning performance, as we discussed in the paper for Random Label and SalUn.
>
> This finding leads directly to efficiency improvements during unlearning. The exact gains are hard to quantify, but one should reasonably expect the efficiency gains to be analogous to the ratio of the size of the low-memorized examples over the overall forget set size.
>
> 3) Our results show that often much more efficient unlearning algorithms (like GA and other similar baselines that mainly need access to the examples in the drastically smaller forget set and do not depend heavily on the drastically larger retain set) can be competitive in terms of ToW/MIA performance for at least some partition (low/medium/high) of the forget set. Thus, for these partitions of the forget set, with RUM one could enjoy much higher efficiency without sacrificing ToW/MIA performance. This will obviously reduce the overall time for unlearning.
>
> 4) In addition, RUM is also shown to be able to significantly improve the performance of all algorithms studied in terms of ToW/MIA. Hence, consider a practitioner who chooses any algorithm (which e.g., for efficiency reasons may not be the best performing in terms of ToW/MIA). By “RUMifying” it, it also boosts its ToW/MIA performance, having a more acceptable and efficient unlearning solution.
>
> We will gladly reflect these discussions about efficiency issues with respect to our contributions in the paper, since, as HxTF points out these are very important (and strengthen our paper).
>
> **Re: the tables and figure are small**: We agree! This is actually an important point. We will address this, making figures and table more readable.
>
> **Re: nit comments on math readability**: great point, we will make this adjustment (and thanks for catching the typo re: the style of S).
>
> **Overall**, we thank the reviewer for the thoughtful feedback and encouraging remarks. We believe that  the discussion above addresses all key elements of efficiency which pertain to our contributions and that our contributions can lead to an efficient and practical deployment (in addition to their contributions to unlearning science). We would love to hear the reviewer’s thoughts on this and discuss further if there are remaining questions. Thanks!

---

> > ### Comment · Reviewer_HxTF · 2024-08-11
> > **Response to rebuttal, other reviewers**
> >
> > Thank you for your thoughtful rebuttal. Your comments and results on efficiency address the majority of my concerns. Thank you for providing these details, I think they make for a stronger paper.
> >
> > I believe that this paper clears the bar for acceptance. I maintain my belief that a score of 7 is appropriate, since I think the work would make for a great poster. I have not assigned a higher score for this reason, but note that I am willing to fight for this paper's acceptance (if necessary, though it seems like it is not tending in that direction).
> >
> > I would to address some comments from other reviewers, which the authors have also addressed. I want to do so in support of their rebuttal. Forgive me if this is, as a result, a bit repetitive. I want to show places where I agree and disagree.
> >
> > *Reviewer Bp7U*
> >
> > > Despite the significance of the insights about sample embeddings, they appear somewhat limited to classification models, which typically have an embedding space to group different samples. While this may be beyond the scope of this paper, it would be valuable to explore extending these investigations to generative models, such as diffusion models and transformer-based large language models (LLMs).
> >
> > Machine learning remains about much more than LLMs. While I think the authors have provided a good rebuttal about doing unlearning on classification as being more generally useful (and while I understand why they may have engaged in a rebuttal like this), I don't think this was necessary. Having great results on classification is a contribution on its own. This paper doesn't need to touch LLMs to make useful contributions.
> >
> > >  The experiments were conducted only on two CIFAR datasets, both involving image classification tasks. Including other tasks or datasets, such as text sentiment or topic classification, would make the paper more comprehensive.
> >
> > I want to support what the authors have said about this. Having done work in this area, it is tremendously expensive. (Related concerns can also be read in my own questions about efficiency -- this type of work is really expensive to do, so it's good to report on this. It also explains why experiments might be done on CIFAR, as opposed to other things). This probably should be listed as a limitation of the work, but is not in my opinion grounds for rejection.
> >
> >  *Reviewer 3Cxb*
> >
> > I think that this reviewer raises important methodological questions about how machine unlearning success is measured in general in the field. In my opinion, the authors have done a good job of contextualizing challenges in the field in their rebuttal, in relation to the presented work. Such discussion is great for the field, and perhaps should go in the Appendix. Nevertheless, based on the comparisons and work done here, I still think that this paper pushes forward the state-of-the-art -- in a field for which there remains a lot of debate about how to measure success for unlearning.
> >
> > That is, while Reviewer 3Cxb raises some excellent points, I do think some of them read as critiques of uncertainty in the field of unlearning, and apply more generally than they do for this specific paper. This paper, in the allotted 9 pages, does a lot to try to improve rigor on measuring unlearning. It is, in my opinion, serving to provide more rigor in this still relatively unclear space. This in my mind serves to improve some of the challenges of unlearning, as articulated by this reviewer.
> >
> > Per-example forgetting results would be interesting and useful, but in my mind are not a blocker for publication of the existing work.
> >
> >  *Reviewer 7Yis*
> >
> > I agree with the assessment that this paper ultimately helps clarify what makes unlearning hard perhaps more than what to do about it. I similarly think this is a sufficient contribution for publication as a poster.

---

> > > ### Comment · Reviewer_3Cxb · 2024-08-11
> > > **Adding onto between-reviewer discussion**
> > >
> > > Thanks HxTF for tying our responses together. Quick note:
> > >
> > > >Per-example forgetting results would be interesting and useful, but in my mind are not a blocker for publication of the existing work.
> > >
> > > I agree with your assessment of my original review, and following the discussion with the author, I am convinced that the paper does make a reasonable contribution on the methods front (probably improving MIA by 3-5 points following their analysis + new method). My biggest concern remains with the example-level comparison. I basically think that aggregating accuracy across datapoints before comparing between models is clearly flawed, since the ToW metric could look good for models that produce very different distributions in practice (and I think pairs of neural models are often like this in practice). The authors convince me that this doesn't totally undermine their contribution, and I raised my score correspondingly, but I would strongly recommend to sort out this issue with the metric or otherwise provide substantial discussion and results in the appendix if they choose to keep the original formulation in the main body of the paper.

---

> > > ### Author Response · Authors · 2024-08-11
> > >
> > > Dear reviewer HxTF:
> > >
> > > Even though you are not raising your score (as we had hoped), we sincerely appreciate your time, effort, and comments -especially as you tried to tie many together. It is gratifying to see reviewers with this level of dedication and devotion to scientific work!
> > >
> > > Thanks a lot!

---

> > > > ### Comment · Reviewer_HxTF · 2024-08-11
> > > >
> > > > I will raise my score if this becomes necessary as a matter of record to advocate for accepting this paper. I hope this is not the case in discussion with the AC, since I think a 7 is fair (I personally would save 8 for the very small % of papers that warrant spotlights/orals).

---

### Official Review · Reviewer_3Cxb · 2024-07-12

**Soundness:** 2
**Presentation:** 4
**Contribution:** 2
**Rating:** 5
**Confidence:** 3

**Summary:**

This paper studies how two properties of “forget sets” make machine unlearning hard: similarity/entanglement between the forget set and an accompanying retain set, and a memorization score for the forget set. The authors hypothesize and experimentally confirm that (1) high entanglement makes unlearning harder, likely due to it being hard to learn a model that forgets data that is so close to retain data, and that (2) more strongly memorized data is harder to forget, because models that memorize such data tend to be quite different from models than models that never see that data. Building on these observations, the authors propose a new algorithm for unlearning that first rank-orders forget data in terms of memorization scores, then selects an unlearning algorithm from a pool of algorithms that achieves the best ToW scores on data with the relevant level of memorization (as determined by previous analysis experiments). Results suggest that sequential forgetting based on forget set memorization achieves immediate improvements over whole-set forgetting, and that leveraging unlearning algorithms with good known ToW scores also improves results.

**Strengths:**

- Very important: The paper discovers a relationship between two retain set properties and unlearning difficulty, and leverages this relationship to improve unlearning performance on some metrics in a setting where these properties can be measured for forget sets. These are interesting findings in their own right, and provide a good starting point for follow up work.
- Important: Experiments are straightforward and easy to understand, with only a few exceptions.
- Important: The paper is very well written and enjoyable to read.
- Important: The paper is well situated in the literature by the introduction, preliminaries, and related work sections. The connection to work on forgetting during training is interesting.
- Of some importance: The paper contains some individual observations that are interesting, though not well supported by the remainder of the analysis. For example, that “operating sequentially on homogenized subsets according to memorization can boost the performance of unlearning algorithms” and that “a key observation from our results is that we can do well by actually doing nothing on a subset of the forget set”.

**Weaknesses:**

- Important: The ToW metric seems very reasonable at first glance but has some worrying issues on further thought. Mainly, the fact that this metric compares average accuracies across a data distribution is problematic. Consider that two models could disagree on many individual data predictions while achieving the same overall accuracy. Thus, an unlearning algorithm could yield a very different model from $\theta^r$ while still achieving good ToW. I like that the metric balances three different desirable quantities, similar to a harmonic mean, but it seems better to balance existing, accepted metrics from the literature, like empirical measurements of divergence between unlearning/original model predictions, MIA, extraction attack success, and accuracy on retain and held-out test sets. Analysis in the paper is only conducted in terms of ToW (Sec 4).
- Important: Results on MIA could be stronger. It appears that the refinement step accounts for most/all of the gains here, which is basically unexpected given the previous analysis in the paper. Improvements on ToW are positive, which would be good for the paper, if I didn’t have some skepticism around the ToW metric described previously.
- Important: Conducting all experiments with ResNet models on CIFAR datasets limits the generalizability of the results. While a breadth of unlearning methods are considered, I think evaluations should be conducted with other datasets and architectures too to ensure generalizability of the results.
- Of some importance: This is a personal opinion: I find the focus on unlearning in the differential privacy vein to be the wrong approach for many unlearning settings. I think it makes sense for users requesting their data not be used in training, and for model developers assuring approximate compliance with this request. I think it doesn’t make sense for unlearning private or dangerous information from models. The paper mentions extraction attacks once in the related work section, but extractability seems like it should be the core metric for unlearning to me. We either want a model to get a datapoint correct or get the datapoint wrong. Whether or not it was used for training seems inconsequential much of the time (one exception is when model developers owe people money for using their data). Reading this paper, this tension appears often. For instance, lines 242-246 reveal that unlearning algorithms can get worse ToW scores when they lower accuracy on a forget set too much, echoed in L.252-253 (cf. “overforgetting”). How can you overforget a forget set? I don’t like this problem formulation. I prefer defining forget sets as data where we want extraction attacks to fail, with retain sets being data where we want high model accuracy, and whether MIAs work on retain data is a matter of user data *access* (not privacy per se, since low-mem data might be relabeled accurately by models with high confidence, even when users do not grant access for training). All this said, I recognize that DP-based definitions for privacy remain popular in unlearning research, so I factor this point into my final score with less weight.

**Questions:**

- Since the retain set is like training data, do you think it would be better to exclude it from ToW?
- Entanglement score: there might be variance in a direction that seems irrelevant to whether the two distributions overlap. I feel that some MMD or classifier-based score could be better here. What do you think?
- In Fig1b, why is Original so low for medium and high memorization retain sets?
- typo: ln171: will cause accidentally erasing

**Limitations:**

I think it is becoming increasingly important to evaluate methods on settings beyond CIFAR+ResNet. This setting is becoming a bit dated compared to current applications of CV or other ML methods. Of course you can’t evaluate on everything, but if the experimental setting is this specific, it could be noted.

On broader impacts, a current (negative) dual use scenario for unlearning techniques is to remove the influence of important safety data from finetuned models, e.g. https://arxiv.org/pdf/2310.03693. I think unlearning work should generally be aware of this scenario.

---

> ### Author Rebuttal · Authors · 2024-08-06
>
> **Re: Weaknesses of ToW**, we discern two interesting issues raised: 1) using average accuracies, 2) that it might be preferable to base ToW on another metric like MIA.
>
> For 1), we computed for each example the output of the unlearned and retrained models, and we count the number of different predictions made by the two models. We find that in all cases there is only a very small percentage of disagreements and that for highly memorized data (which generally makes unlearning harder) the number of disagreements increases, but remains small. For low-memorized data, across all algorithms tested, the percentage of disagreements is within ca. [3%, 5%], while for highly-memorized data, it ranges between ca. [7%, 15%]. We will add these results to the paper.
>
> For 2) we note:
> - There is an ongoing debate for the most appropriate metric for unlearning. An advantage of ToW is the possibility to compare unlearning algorithms “holistically” based on a single scalar.
>
> - ToW does use existing metrics: the “accuracy gap” (between the unlearned and retrained models), is commonly used in the literature!
>
> - While we used ToW for our analyses (for simplicity and efficiency), we did evaluate RUM with MIA too. The fact that RUM improves w.r.t. MIA as well can be seen as an indication that ToW is actually a useful proxy for analyzing factors affecting difficulty that generalizes to other metrics.
>
> - But to address this more concretely, we tried a variant coined “ToW-MIA” that replaces the accuracy gap on the forget set term with the MIA gap (see Section 6). The results shown in Figures 7-9 in the PDF clearly show the same trends with ToW-MIA as for ToW, both for analyses and RUM. We will add these to the paper.
>
> **Re: “Results on MIA could be stronger” and “ refinement step accounts for most/all gains”**
> - Our results in general show that both the refinement and per-subset algorithm choice of RUM play a big role. We do not see why it is “unexpected” to see that, for MIA, the refinement step plays a bigger role. This is actually a nice finding, which may lead to further insights.
>
> - Table 1 in the paper clearly shows that RUM introduces strong MIA gains! For any algorithm there, the RUM version has much better MIA (and ToW). We are thrilled with this result (and frankly surprised to read this criticism).
>
> **Re: limited to ResNet and CIFAR**. We address this point by adding an additional dataset and architecture. Please see the “common response” to all reviewers. We hope this considerably helps to put at ease doubts about the generalizability of our results.
>
> **Re: personal opinion on unlearning, privacy and “whether or not it was used for training seems inconsequential”**: We believe different applications require different treatment. Here we are addressing unlearning defined as removing a subset of training data (the forget set). We agree some problems (e.g. preventing harmful generations) are not necessarily formulated as this type of “unlearning” because, as the reviewer said, all we care about there is to prevent those “bad” outputs (regardless of what data was used for training). In that case, retraining from scratch isn’t even the right reference point. However, our formulation is useful in other cases, e.g. enabling data deletion from a vision classifier. In that case, it is reasonable (and common) to adopt this DP-like formulation: ideal “deletion” of a data point leads to a model similar to one that was trained without that point. And in this case, MIAs are a common and convenient tool for measuring “leakage” and success of this type of unlearning. Overall, these are highly debated topics, with subjective opinions, and without consensus. While work is needed to settle these debates, we hope the reviewer agrees beyond doubt that our problem setting is widely considered a valuable one.
>
> **Re: “how can you overforget a forget set?”** Following from above, our goal is to produce an unlearned model “similar” to that of retrain-from-scratch. So its accuracy on the forget set should be just as low as that of retraining, but no lower. We use the term “overforgetting” to describe the situation where the unlearned model causes a forget set accuracy lower than that of retrain. This is problematic as it can lead to vulnerability to MIA (due to noticeable differences from retraining).
>
> **Re: should we exclude retain set from ToW?** The retain set is important. For instance, if a user has contributed some data towards training a model, and that user is not in the forget set, the user wants to still benefit from the model performing well for their data. At the same time, we want to preserve the ability to generalize to new users (test performance).
>
> **Re: using MMD instead of ES**: Interesting point! We experimented with MMD and find there exists a (negative) correlation between ES and MMD scores. For CIFAR10/ResNet18 and CIFAR100/ResNet50, there exists a negative correlation where low(high) ES scores lead to high (low) MMD values.
>
> **Re: “why is original so low for medium and high memorization retain sets”**. We assume the reviewer means “forget sets” rather than “retain sets” here (please correct us if not). An example that is “low mem” is one that has not influenced the model too much by definition (see Equation 1). So the predictions of the “original model” already look like those of the “retrained” model for this example, making the original model represent “good unlearning” according to the metrics of interest (based on similarity to the retrained model). On the other hand, for higher-mem examples, the original model is more influenced by those examples and therefore it “looks less” like a model that was trained without those examples. In these cases, the original model is a poor solution. Please let us know if this clarification helps.
>
> **Re: limitations**, we will note these, thank you.
>
> We hope our rebuttal satisfies all key concerns of Reviewer 3Cxb. We look forward to discussing any remaining concerns.

---

> > ### Comment · Reviewer_3Cxb · 2024-08-09
> > **Reply to rebuttal**
> >
> > Thanks for the response! Replies below.
> >
> > > **Re: Weaknesses of ToW**, we discern two interesting issues raised: 1) using average accuracies, 2) that it might be preferable to base ToW on another metric like MIA.
> >
> > Thanks for the response to (1). I would recommend adjusting the core metric to avoid any possible bias in results. If I’m reading the response correctly, it seems like a measure of bias could be as high as 15%. For (2), I think ToW-MIA is a good idea and appreciate the additional results.
> >
> > > We do not see why it is “unexpected” to see that, for MIA, the refinement step plays a bigger role. This is actually a nice finding, which may lead to further insights.
> >
> > To clarify, I meant unexpected here as in, the paper is focused on how two factors could explain unlearning difficulty, and then the results show that a third and seemingly arbitrary factor (the ordering of data into subsets based on memorization scores) plays the biggest explanatory role. The bulk of the paper does not concern data ordering. Why is data ordering important for unlearning? A whole different paper could be written on this (the paper points to [5] for relevant work).
> >
> > > Table 1 in the paper clearly shows that RUM introduces strong MIA gains! For any algorithm there, the RUM version has much better MIA (and ToW). We are thrilled with this result (and frankly surprised to read this criticism).
> >
> > To clarify, I was looking at MIA falling from 0.089 (L1 vanilla) to 0.059 (RUM). Is this a strong gain (and is it statistically significant)? Is this the comparison I should be looking at?
> >
> > > **Re: limited to ResNet and CIFAR**. We address this point by adding an additional dataset and architecture.
> >
> > Adding VGG-16 with Tiny ImageNet is an improvement
> >
> >  >….MIAs are a common and convenient tool for measuring “leakage” and success of this type of unlearning. Overall, these are highly debated topics, with subjective opinions, and without consensus. While work is needed to settle these debates, we hope the reviewer agrees beyond doubt that our problem setting is widely considered a valuable one.
> >
> > I agree. It’s a minor point I wanted to make. I suspect that as the public learns more about ML + data privacy/deletion, there will be a shift towards extraction attack metrics over DP metrics, but I can’t deny that this is a legitimate and longstanding problem setting.
> >
> > > **Re: “how can you overforget a forget set?”** Following from above, our goal is to produce an unlearned model “similar” to that of retrain-from-scratch. So its accuracy on the forget set should be just as low as that of retraining, but no lower. We use the term “overforgetting” to describe the situation where the unlearned model causes a forget set accuracy lower than that of retrain. This is problematic as it can lead to vulnerability to MIA (due to noticeable differences from retraining).
> >
> > Thanks for the clarification. The argument makes sense to me. I get the feeling the terminology in the field (not just in the paper) could be tightened up a bit, because it is a little unintuitive to speak of “overforgetting” a forget set. (Forget sets should be forgotten!)
> >
> > > **Re: should we exclude retain set from ToW?** The retain set is important. For instance...
> >
> > Right, that makes sense.
> >
> > > We experimented with MMD and find there exists a (negative) correlation between ES and MMD scores.
> >
> > Thanks!
> >
> > > **Re: “why is original so low for medium and high memorization retain sets”**...Please let us know if this clarification helps.
> >
> > Got it, thanks!
> >
> > ---
> >
> > Based on the above discussion, I am prepared to increase my score if the authors can (1) reformulate the ToW metric to measure example-level changes in model predictions and not changes in average accuracy alone, and (2) clarify the strength of evidence there is for improved MIA.
> >
> > I would still point out that there is something confusing to me about this paper regarding the refinement step. The analysis in the paper focuses on memorization and entanglement, not data ordering. But then data ordering turns out to be very important for the improvements in unlearning in the proposed method.

---

> > > ### Author Response · Authors · 2024-08-09
> > > **Answers to new questions of reviewer 3Cxb**
> > >
> > > **“Re: …adjusting the core metric to avoid … bias. …  it seems like a measure of bias could be as high as 15%.”**
> > >
> > > We created ToW as a reasonable metric that holistically captures a notion of "difficulty" in a scalar, **to allow for efficient analyses and investigations into unlearning difficulty**. The reviewer’s point about looking at disagreements at the example level rather than average accuracy is a great one! As far as our goals were concerned with respect to using ToW as a tool to measure unlearning difficulty, we actually see the same trends with our initial framing of ToW compared to this example-based disagreement score!
> > >
> > > Concretely, for the vast majority of unlearning algorithms, we find according to ToW that low mem forget sets are easier. Correspondingly, we also observe that for low mem examples, the disagreement rate is also lower. On the other hand, higher mem examples are harder to forget, as shown by average accuracy (in original ToW) and indeed the disagreement rate is also higher.
> > > Given that our goal is to study trends, and we see the same trends emerging according to both average accuracies and example-level disagreements, we feel there is no issues of “bias”; we in fact view this agreement of trends as evidence for the robustness of conclusions one would draw when using our contributions.
> > > Curious to hear the reviewer’s thoughts on this.
> > >
> > > **Re: “To clarify, I was looking at MIA falling from 0.089 (L1 vanilla) to 0.059 (RUM). Is this a strong gain (and is it statistically significant)? Is this the comparison I should be looking at?”**
> > >
> > > We think the comparison that makes sense is to compare L1-sparse vanilla against L1-sparse RUM. Maybe accidentally the reviewer looked at the wrong rows in the table? (comparing L1-sparse vanilla against NegGrad+ RUM, but please correct us if we misunderstood your comparison).
> > >
> > > In brief, yes they are statistically significant. For L1-sparse in Table 1, the MIA gap falls from 0.089 to 0.033. Specifically, we computed the paired t-test to derive the CIs for the MIA gaps. For the MIA gap for (L1-sparse-vanilla - Retrain) the 95% CI is 0.089 +/- 0.032. For the MIA gap for (L1-sparse-RUM - Retrain) the 95% CI is 0.033 +/- 0.003.
> > >
> > > **Re: “the paper is focused on how two factors could explain unlearning difficulty”, “a seemingly arbitrary factor (the ordering of data [...]) plays the biggest explanatory role.” and importance of data ordering.**
> > >
> > > Thank you for the discussion! What we show with RUM is that operating on **homogenized subsets of the forget set** (obtained in the refinement step) yields big gains (over random subsets as in our Shuffle baseline), and further, that pairing each such subset with the most suitable unlearning algorithm yields further gains. The factors explaining difficulty (that we discover through analyses in Section 4) are what informs both the “refinement”, i.e. how we divide the forget sets into subsets (using memorization levels), as well as how to pick the most suitable unlearning algorithm per subset.
> > >
> > > Now, RUM (as presented in the paper) makes the design decision of operating sequentially on refined subsets, which introduces this additional variable of the ordering, as the reviewer points out.
> > > Our work shows **the existence of orderings** that can yield large gains in unlearning pipelines. We hope future work extends our work to yield even larger gains in RUM, including a thorough investigation of the data ordering which is a very interesting component (but well beyond the scope of this work).
> > >
> > > **Overall and proposed adjustments to the paper:**
> > > We sincerely wish to thank Reviewer 3Cxb: (i) for engaging fruitfully and thoroughly with us and (ii)  for their insightful comments that have already improved our paper.
> > >
> > > We are happy that 3Cxb appreciates the results with ToW-MIA, the example disagreement experiments, adding TinyImageNet and VGG-16 and acknowledges that, while unlearning definitions and subproblems may differ, the variant we study here is widely recognized as valuable.
> > >
> > > Taking reviewer 3Cxb’s comments to heart, we propose the following: When introducing ToW, we will clarify our motivation behind this metric (studying difficulty via a reasonable and informative scalar value), but we will also discuss variations of it in two axes: (i) per-example disagreement rate (unlearning vs retraining) in addition to average accuracy, and (ii) defining ToW-MIA (using MIA gap for forgetting quality, instead of forget accuracies). We will add sections in the Appendix to each of those points, showing the results of additional investigation in response to the reviewer’s feedback. We note that all of these variants show the same trends, which we will also note in the main paper, and we believe supports the robustness of our conclusions further.
> > > We thank the reviewer for bringing up these points which (we had not thought of and) strengthen our work.
> > >
> > > Please let us know if you would require anything else?
> > >
> > > Again, thanks!

---

> ### Comment · Reviewer_3Cxb · 2024-08-11
> **Additional reply to rebuttal**
>
> >The reviewer’s point about looking at disagreements at the example level rather than average accuracy is a great one…and we see the same trends emerging according to both average accuracies and example-level disagreements, we feel there is no issues of “bias”; we in fact view this agreement of trends as evidence for the robustness of conclusions one would draw when using our contributions. Curious to hear the reviewer’s thoughts on this.
>
> Sure, so it seems like we agree the example level metric would look very different from the dataset level metric. In the limit, accuracy might not change at all while the two models might disagree on most examples, by virtue of trading off similar amounts of false positives vs false negatives.
>
> So, the metric is flawed at its core. It is very possible that two models get similar performance while producing very different predictive distributions. In fact this is common for two comparable NNets trained on a dataset, when the models don’t get near 100% accuracy. The metric would suggest that these models are indistinguishable, when this is not the case.
>
> The fact that, for the few models and datasets examined in this paper, this bias does not seem to exceed 15% (a large number) is not comforting to me. We know in theory it’s wrong. We should just change it.
>
> Note that this example-level disagreement can still become a scalar metric. It being a scalar metric is not the crux here.
>
> >We think the comparison that makes sense is to compare L1-sparse vanilla against L1-sparse RUM. Maybe accidentally the reviewer looked at the wrong rows in the table? (comparing L1-sparse vanilla against NegGrad+ RUM, but please correct us if we misunderstood your comparison).
>
> Well, it’s not a “wrong” comparison. There are two ways of doing the comparison. You could control for the underlying method (L1-sparse vs NegGrad) and assess the effect of RUM across such settings. Or you could ask, how good can RUM be across methods, compared to how well we can do across methods without RUM. Two different, valid comparisons. And for what it’s worth, always restricting to within-method comparisons suffers from the risk of multiple testing (https://en.wikipedia.org/wiki/Bonferroni_correction).
>
> > In brief, yes they are statistically significant. For L1-sparse in Table 1, the MIA gap falls from 0.089 to 0.033. Specifically, we computed the paired t-test to derive the CIs for the MIA gaps. For the MIA gap for (L1-sparse-vanilla - Retrain) the 95% CI is 0.089 +/- 0.032. For the MIA gap for (L1-sparse-RUM - Retrain) the 95% CI is 0.033 +/- 0.003.
>
> Great! I would recommend including this in the paper. In short, I am willing to believe there is an improvement from RUM, on the order of 3-5 points MIA.
>
> > The factors explaining difficulty (that we discover through analyses in Section 4) are what informs both the “refinement”,
>
> Ok, this is a good point. Doing the subsetting/ordering depends on identifying the factors in the first place.
>
> ___
>
> Thanks for the proposed changes. I think they seem adequate. I would overhaul the metric, but if it’s the case that a reader immediately sees a footnote or appendix reference clarifying the average vs. example level difference (and results are also provided in the appendix with an example level metric), that’s ok. I leave it to the authors to include this discussion in the revised manuscript, and I will raise my score to Borderline Accept.

---

> > ### Author Response · Authors · 2024-08-11
> >
> > Dear reviewer 3Xcb
> >
> > Thank you for your comments.
> > We assure you we will provide the additional discussion on the issues pertaining to example-level performance and our relevant results. We view this as an issue going beyond our paper - an important issue for future related work to consider going forward, thus affecting the community's research effort.
> >
> > Thanks!

---

### Official Review · Reviewer_Bp7U · 2024-07-13

**Soundness:** 3
**Presentation:** 2
**Contribution:** 3
**Rating:** 5
**Confidence:** 3

**Summary:**

This paper investigates the characteristics of forget data for machine unlearning algorithms and demonstrates their impact on the effectiveness of various unlearning methods. The paper further proposes refining the forget set into subsets based on data characteristics and selecting suitable unlearning algorithms for each subset. The experimental results show that this ensemble method outperforms single baselines.

**Strengths:**

* The identification of properties in forgetting data is valuable for machine unlearning. The idea of adapting algorithms to the properties of specific data could benefit future algorithm design.
* The proposed refine-based method is simple yet effective, as demonstrated by the experiments.

**Weaknesses:**

* Despite the significance of the insights about sample embeddings, they appear somewhat limited to classification models, which typically have an embedding space to group different samples. While this may be beyond the scope of this paper, it would be valuable to explore extending these investigations to generative models, such as diffusion models and transformer-based large language models (LLMs).
* The experiments were conducted only on two CIFAR datasets, both involving image classification tasks. Including other tasks or datasets, such as text sentiment or topic classification, would make the paper more comprehensive.
* Although the paper claims the RUM framework is a meta-algorithm framework, the selection of unlearning algorithms for each subset is now based on empirical investigation and human-designed heuristic, which limits the generalizability of this method.

minor: I would recommend refining Figure 2, as some text is misaligned and mixed with the box boundaries.

**Questions:**

* What is the model accuracy before unlearning in experiments?
* What's the criteria to group samples into low/medium/high partitions?

**Limitations:**

Authors discussed the limitations.

---

> ### Author Rebuttal · Authors · 2024-08-06
>
> **Re: limited to classification models.** We agree it is important to study different modalities, discriminative and generative models. However, these settings differ in three key ways.
>
> 1) Unlearning definitions: The problem we study is removing the influence of a particular subset of training data (the “forget set”) from the model; success is defined as matching the distribution of retraining from scratch without the forget set. In generative models, unlearning can take different forms: i) remove a ‘concept’ (e.g. nudity, in diffusion models; Fan et al), or harmful content in LLMs, or ii) in LLMs, unlearn memorized text. In both i and ii, the goal is not matching retrain-from-scratch. For case i, the “forget set” may not even contain training images, it may be generated instantiations of unwanted content, and application-dependent metrics are used. Though ii is closer to our problem, computing retrain-from-scratch is not practical in large LLMs (none of the works use that as a reference point) and other definitions or metrics are used.
>
> 2) Memorization definitions. According to the definition of label memorization (see equation 1), rare or atypical examples are more highly memorized, as we discussed. Instead, in LLMs, memorization is typically defined as “verbatim memorization” (e.g. see Definition 2 in [1]), capturing the model’s ability to regurgitate training data. Recent works (e.g. [1]) find that the more common a string is (i.e. less rare), the more likely it is to get memorized. This definition is what SOTA methods for memorized-data unlearning [2,3] use (case ii in the above). These notions are clearly distinct (and even seemingly at odds with one another) and an analysis of unlearning difficulty for verbatim memorization would thus require specialized treatment.
>
> 3) SOTA methods. For example, the SOTA models for unlearning memorized data from LLMs are based on “simple” Gradient Ascent (GA) (see [2,3]). On the contrary, in computer vision / image classification tasks, (variations of) GA have been shown to be lacking badly. In particular, model utility for image classification suffers largely with GA (and variations thereof), whereas for text generation and reasoning, model utility can remain high with GA (as shown in for example [2,3].
>
> References: [1] Carlini et al, 2021 “extracting training data from large language models”, [2] Jang et al. 2023. “Knowledge unlearning for mitigating privacy risks [...]”, [3] Barbulescu et al. 2024. “To each (textual sequence) its own [...]”.
>
> In summary, unlearning in generative models differs fundamentally. Hence, as the reviewer states, a comprehensive analysis is (well) beyond the scope of this paper. Our work is an important step in advancing knowledge and in-depth understanding of unlearning for image classification.
>
> **Re: only CIFAR and image classification.**
> We have added a new dataset and architecture based on this feedback, please see the “common response” for the details. All conclusions remain consistent, supporting the generalizability of our findings. We hope this addresses the reviewer’s concern.
>
> Please also note our answer to the above point for the comment on text generation and reasoning. We leave an investigation of tasks in other modalities and generative models for future work for those reasons.
>
> **Re: “human-designed heuristic [...] limits the generalizability of this method.”**
> Thank you for raising this, we believe this is a very important point.
>
> First, our primary goal is a deeper understanding of behaviours of unlearning methods on different forget sets.  Crucially, our work shows the potential of such insights leading to better unlearning pipelines: each SOTA algorithm is shown to be improved upon using RUM. And, RUM using different algorithms for different partitions further improves performance. This is already a key contribution that can inform better generalizable approaches in the future.
>
> Of course, repeating our entire analysis on each new dataset or problem setting in order to plug in the “hand-picked” design decisions when using RUM is undesirable. Due to this, we designed our analysis to uncover key factors affecting algorithm behaviours (e.g. how algorithms interact with memorization) that transfer across datasets / architectures (e.g. as long as one knows the memorization scores). Since computing memorization scores for new training datasets is expensive, we show in our new experiments results with a cheap proxy (the “C-proxy”); see the “common response”. Hence, RUM is generalizable / practical: First, uncover key characteristics, for instance, memorization levels (or a cheaper proxy), and then plug them in to RUM to get big performance gains.
>
> **Re: minor**: thank you, we will fix these.
>
> **Re: “What is the model accuracy before unlearning in experiments”** – we denote this by “Original” in each experiment. The test accuracy values are 84.353±3.455% for CIFAR-10/ResNet-18 and 75.003±2.809% for CIFAR-100/ResNet-50, with 100% training accuracy for both settings. More detailed accuracy results can be found in the appendix: Table 5,6 for unlearning difficulty vs ES experiments, and Table 8,9 for unlearning difficulty vs memorization experiments.
>
> **Re: “What is the criteria to group samples into low/medium/high partitions”**. First, we sort in increasing order all examples in the dataset based on the metric of interest (memorization score or embedding distance). Then, say the forget-set size is N. We select the first (lowest value) N/3 examples to be the ‘low partition’, the last N/3 examples to be the ‘high partition’ and the middle N/3 to be the ‘medium partition’.
>
> **Overall**, we thank the reviewer for the thoughtful critical feedback. We believe the new results (with C-proxy and additional dataset / architecture) and discussion, address the reviewers central concerns. We would love to hear the reviewer’s thoughts and discuss further if there are remaining concerns. Thanks!

---

> > ### Comment · Reviewer_Bp7U · 2024-08-09
> >
> > Thanks for the detailed reply, I have read the rebuttal. My major concern about the efficiency is resolved by the new C-proxy method. I would maintain the current positive rating.

---

> > > ### Author Response · Authors · 2024-08-10
> > >
> > > Thank you for engaging with us.
> > >
> > > We are glad to see that we have addressed your concerns regarding efficiency.
> > >
> > > We are very keen to engage further to resolve your other remaining concerns (regarding our claim that unlearning in LLMs is a substantially different problem with different definitions for the problem itself and for key concepts like memorization, and with different SOTA algorithms).
> > >
> > > Please do get in touch if there is anything else we can do to help alleviate any remaining concerns.
> > >
> > > Thanks again.

---

### Author Rebuttal · Authors · 2024-08-06

We thank all reviewers for the valuable feedback. We are pleased to hear that the reviewers found our analysis “valuable”, that we study “very sensible questions”, “very comprehensively”, and our paper “makes excellent contributions to the unlearning science”; that our findings are good both “in their own right” as well as “for follow-up work”, the paper “well-written” and “well-situated in the literature”.

In this general response, we address concerns shared by more than one reviewer. We respond to individual concerns in the per-reviewer responses separately.

**Re: practicality and efficiency.** To address reviewers concerns on the efficiency and practicality of the RUM framework, in particular as it relates to the cost of computing memorization scores, we experimented with a proxy for these scores, that we coin as the C-proxy, based on the confidences, as proposed in [1] (see reference below). Specifically, for a data point (x,y) and a model M, we measure the softmax confidence of M(x) corresponding to the correct class y at each epoch during the training of the original model. The C-proxy metric is then computed by averaging these confidences across all training epochs for each data point at the end of the training process. The C-proxy exhibits a strong negative correlation with memorization, with a Spearman correlation coefficient of approximately -0.9.

We begin by showing (in Figures 1 and 2 of our uploaded PDF) that C-proxy behaves the same as using memorization levels themselves (by contrasting these Figures with the corresponding figures in our paper). Based on this observation, we conduct other rebuttal experiments (e.g. see the below point for other datasets and architectures) using this C-proxy, since it’s more efficient.

Please note that computing C-proxy instead of Feldman’s memorization score enjoys orders of magnitude performance improvement (due to avoiding training a large number of models that are needed to compute Feldman’s memorization scores). This makes our RUM framework practical and efficient to deploy.

**Re: we only used two dataset-architecture pairs (CIFAR-10 with ResNet-18 and CIFAR-100 with ResNet-50)**

First, we would like to emphasize that producing the results and deriving the insights in the paper was enormous amount of effort. Specifically, training different size models, several times (in order to e.g. compute memorization scores) across different datasets, using many state of the art algorithms and analyzing them with respect to ToW and MIA as they are affected by both memorization scores and embedding space distances of forget-set examples takes enormous effort, as we are the first to explore this space. And in addition to the above effort we further showed the effectiveness of RUM, for different combinations of SOTA and baseline algorithms and across datasets and architectures. Please note as well that, thanks to our analyses (and also the discovery that C-proxy yields similar findings as computing memorization scores), future efforts can be much more computationally-efficient thanks to this effort. We will now present evidence in support of RUM’s generalizability to new datasets/architectures as well.

We agree that using only two CIFAR datasets is a valid concern. To respond to it, we provide results both: a) with an additional dataset (Tiny-imagenet) using Resnet18, and b) with Tiny-imagenet using an additional architecture (VGG16), and we do this both for the analyses in Section 4 (investigating behaviours of algorithms w.r.t ES and memorization) and in the context of RUM.

These results Figures 3-6 in the PDF, agree completely with the previous results in the paper. Specifically, all trends identified with the previous experiments (e.g., regarding the impact of memorization and ES on unlearning difficulty) and the impact of RUM in improving each algorithm in isolation and overall are manifested with the new dataset and with the architecture. We believe this is important evidence for the generalizability of our findings and we hope alleviates this concern.

**Overall**, we thank the reviewers for the insightful comments and believe that we have addressed the concerns of all reviewers during the rebuttal, which has improved our paper. We are eagerly awaiting the reviewers’ responses (on these general points and per-reviewer rebuttals) and we look forward to continue these discussions.

References:
[1] Jiang, Ziheng, et al. "Characterizing Structural Regularities of Labeled Data in Overparameterized Models." International Conference on Machine Learning. PMLR, 2021.

---

### Decision · Program_Chairs · 2024-09-25

**Decision:**

Accept (poster)

**Comment:**

This paper looks at the machine unlearning problem — roughly, given a model m and a subset of the data (the “forget set”) used to train that model, efficiently create a new model m’ that does not include information from the forget set but is representative of the family of models that would’ve been trained on the initial data sans the forget set (the “retain set”).  This is a known tricky problem, and a well-motivated one due to so-called “right to forget” capabilities enforced in the EU’s GDPR and other evolving regulatory movements.  The paper finds that if the forget and retain sets are close, in some sense, then the unlearning problem becomes harder.  It also finds that, if elements of the forget set are fundamentally part of the model itself via memorization, then the unlearning problem becomes harder.  Neither of these results are particularly surprising, but the analysis is strong and this is great to have written down.  The paper then provides a new method based on these two insights that arguably improves unlearning over various SotA and SotA-adjacent methods.  Reviewers found issue with the experimental analysis, as well as some nitpicks around the identified sources of hardness, but these were largely assuaged in the rebuttal and post-rebuttal discussion.